# Two New Methods Based on Implicit Expressions and Corresponding Predictor-Correctors for Gravity Anomaly Downward Continuation and Their Comparison

Chong Zhang [1,2,3,4], Pengbo Qin [5,*], Qingtian Lü [2], Wenna Zhou [6] and Jiayong Yan [2]

1 State Key Laboratory of Lunar and Planetary Sciences, Macau University of Science and Technology, Macao 999078, China; zchong_chn@163.com or zhangchong@cags.ac.cn
2 Laboratory of Deep Earth Science and Exploration Technology Ministry of Natural Resources (Sinoprobe Laboratory), Chinese Academy of Geological Sciences, Beijing 100037, China; lqt@cags.ac.cn (Q.L.); yanjy@cags.ac.cn (J.Y.)
3 National Engineering Research Center of Offshore Oil and Gas Exploration, Beijing 100028, China
4 Chengdu Institute of Geology and Mineral Resources, Chinese Geological Survey, Chengdu 610041, China
5 Guangzhou Marine Geological Survey, China Geological Survey, Guangzhou 510075, China
6 School of Earth Sciences, Key Laboratory of Mineral Resources in Western China (Gansu Province), Lanzhou University, Lanzhou 730000, China; zhouwn@lzu.edu.cn
* Correspondence: qinpengbo@mail.cgs.gov.cn; Tel.: +86-13318876214

**Abstract:** Downward continuation is a key technique for processing and interpreting gravity anomalies, as it has a major role in reducing values to horizontal planes and identifying small and shallow sources. However, it can be unstable and inaccurate, particularly when continuation depth increases. While the Milne and Adams–Bashforth methods based on numerical solutions of the mean-value theorem have partly addressed these problems, more accurate and realistic methods need to be presented to enhance results. To address these challenges, we present two new methods, Milne–Simpson and Adams–Bashforth–Moulton, based on implicit expressions and their predictor-correctors. We test the validity of the presented methods by applying them to synthetic models and real data, and we obtain stability, accuracy, and large depth (eight times depth intervals) downward continuation. To facilitate wider applications, we use calculated vertical derivatives (of the first order) by the integrated second vertical derivatives (ISVD) method to replace theoretical ones from forward calculations and real ones from observations, obtaining reasonable downward continuations. To further understand the effect of introduced calculation factors, we also compare previous and presented methods under different conditions, such as with purely theoretical gravity anomalies and their vertical derivatives at different heights from forward calculations, calculated gravity anomalies and their vertical derivatives at non-measurement heights above the observation by upward continuation, calculated vertical derivatives of gravity anomalies by the ISVD method at the measurement height, and noise. While the previous Adams–Bashforth method sometimes outperforms the newly presented methods, new methods of the Milne–Simpson predictor-corrector and Adams–Bashforth–Moulton predictor-corrector generally present better downward continuation results compared to previous methods.

**Keywords:** gravity anomaly; downward continuation; numerical solution; mean-value theorem; implicit expression; explicit expression; predictor-corrector

## 1. Introduction

It is significant to improve the accuracy and reliability of results in observed gravity data processing and interpretation for the successful detection of geological structures and exploration of mineral resources [1–5]. In general, data from multiple heights can enrich the description of detected targets and improve the accuracy and reliability of processing and interpretation [6,7]. A variety of gravity data can be obtained from different altitudes such as satellite, airborne, ground (or ocean), and even seabed, to create multiple height datasets [8].

However, it is not always possible to obtain multiple height data simultaneously over the same gravity measurement area in general; for example, traditional ground-based gravity measurements can be difficult and inefficient to implement due to actual observation factors such as dense forests or deserts. Aerial-based gravity measurements using satellites or airbornes can avoid such difficulties, but the observed gravity data decays with the inverse square of the distance from the Earth's center, with the increase in height resulting in the loss of information details [9]. Downward continuation of aerial-based observed gravity data to a lower level, such as the ground, can enhance the resolution [2]. Therefore, a downward continuation of observed gravity data and corresponding anomalies can highlight local and shallow geological information, which plays a critical role in the processing and interpretation of gravity data and anomalies [10] and further improves the accuracy and reliability of mineral resources exploration.

Downward continuation of the gravity anomaly is an ill-posed problem, and general methods cannot achieve stable and accurate results and deep continuation depth simultaneously [10]. The fast Fourier transform (FFT) method is commonly used, but its downward continuation factor amplifies high-frequency components, and its truncation error in the Fourier transform causes oscillations in the continuation procedure [11]. Improved methods, such as the regularization FFT method, make downward continuation stable, but their depths are not large, with typically no more than five times the spacing interval [12]. The integral iteration method is stable, accurate, and has deep continuation depth [13]. However, its iteration number is large and noise in data will be accumulated heavily during iterations. The Milne and Adams–Bashforth methods based on numerical solutions of the mean-value theorem using the observed vertical derivative of the first order, simply called vertical derivative herein, at the measurement height, are reported to be stable, accurate, and have deep continuation depth [14–16]. However, these two methods' truncation errors are not small enough mathematically compared to other methods based on numerical solutions of the mean-value theorem [17].

To develop a more accurate downward continuation method based on numerical solutions of the mean-value theorem with an equally stable procedure and a deep continuation depth, we present two new methods of the implicit expressions and their predictor-correctors for gravity anomaly downward continuation. As the advantages of previous downward continuations based on numerical solution methods are limited by the observed vertical derivatives at the measurement height, we consider wider application scenarios using the calculated vertical derivatives, such as those without observed vertical derivatives, while maintaining stability, accuracy, and deep continuation depth. We introduce the ISVD method [18] to calculate the vertical derivative at the measurement height. The results show that new methods can provide stable, accurate, and deep-depth downward continuations. To further understand the presented methods, we compared and analyzed them by changing introduced calculation factors in downward continuation procedures: upward continuation, vertical derivative calculation, and noise disturbance. Factors, such as calculated gravity anomalies and their vertical derivatives at non-measurement heights above the observation by upward continuation, calculated vertical derivatives of gravity anomalies at the measurement height by the ISVD method, and noise, do affect the four numerical solution-based downward continuation methods, but to different extents. Overall, the downward continuation methods based on numerical solutions of the mean-value theorem perform better than the integral iteration method, and the newly presented Adams–Bashforth–Moulton predictor-corrector method is a better choice than the other three numerical solution-based ones.

## 2. Methods

To understand the methods, we first present general expressions of numerical solutions of the mean-value theorem for gravity anomalies. Second, we recall the explicit Adams–Bashforth and explicit Milne expressions for gravity anomaly downward continuation. Finally, we derive and present two implicit expressions of Adams–Moulton and Simpson

and their predictor-corrector methods of Adams–Bashforth–Moulton and Milne–Simpson for downward continuation.

*2.1. Two Explicit Expressions for Downward Continuation*

2.1.1. Numerical Solutions of the Mean-Value Theorem for Gravity Anomalies

The mean-value theorem for the gravity anomaly can be [14]:

$$g(x,y,z_0+h) \equiv g(x,y,z_0) + g'(x,y,\zeta)h, \ z_0 \le \zeta \le z_0 + h, \tag{1}$$

where $g(x,y,z_0+h)$ is the gravity anomaly at the height of $z_0 + h$, $(h > 0)$, $g(x,y,z_0)$ is the gravity anomaly at the height of $z_0$, $g'(x,y,\zeta)$ is the vertical derivative (of the first order) of the gravity anomaly at the height of $\zeta$, and $\zeta$ is a constant number in $[z_0, z_0 + h]$. The positive $z$ axis is vertically downward throughout the method section.

Numerical solutions can be ways to solve the mean-value theorem of (1), which can be rewritten as a first-order ordinary differential equation [17]. The numerical solutions can be regarded as formulae for the downward continuation of potential fields [14]. One category of numerical solutions is multistep methods [16] and can be generally expressed for gravity anomalies as:

$$\sum_{j=-1}^{s} \alpha_j g(x,y,z_0 - jh) = h \sum_{j=-1}^{s} \beta_j g'(x,y,z_0 - jh), \tag{2}$$

where coefficients $\alpha_j$ and $\beta_j$ are determined by a polynomial passing through the $s$ step solutions, the $s$ steps represent different heights, and the $s$ step solutions represent gravity anomalies at these heights herein. Expression (2) is slightly different in symbols and their meanings from the recurrence relation of multistep methods of Equation 3.41 in [17]. To determine the gravity anomaly of downward continuation at the height of $z + h$, $(h > 0)$, we require $\alpha_{-1} \ne 0$ and assume $\alpha_{-1} = 1$. We obtain the general expression for the gravity anomaly downward continuation by multistep methods of numerical solutions of the mean-value theorem:

$$g(x,y,z_0+h) = -\sum_{j=0}^{s} \alpha_j g(x,y,z_0 - jh) + h \sum_{j=-1}^{s} \beta_j g'(x,y,z_0 - jh). \tag{3}$$

2.1.2. Explicit Adams–Bashforth and Explicit Milne Expressions for Downward Continuation

To utilize these methods, set $\beta_{-1} = 0$ in Equation (3), corresponding gravity anomaly downward continuations are explicit expressions. Skipping the explicit expression of the forward Euler method for downward continuation, which has low accuracy due to its approximate calculation with only two terms ($\alpha_j = -1 \ for \ j = 0$, $\alpha_j = 0 \ for \ j \ne 0$; $\beta_j = 1 \ for \ j = 0$, $\beta_j = 0 \ for \ j \ne 0$), explicit expressions of the Adams–Bashforth and the Milne are tested to be workable for real problems of downward continuation [14–16], and their expressions with the same order of forth [19–21], $k = 4$, can be written for gravity anomaly downward continuationas the explicit fourth-order Adams–Bashforth formula with the error term:

$$\begin{aligned} g(x,y,z_0+h) = \ & g(x,y,z_0) + h\left(\frac{55}{24}g'(x,y,z_0) - \frac{59}{24}g'(x,y,z_0 - h)\right. \\ & \left. + \frac{37}{24}g'(x,y,z_0 - 2h) - \frac{9}{24}g'(x,y,z_0 - 3h)\right) \\ & + \frac{251h^5}{720}g^{(5)}(x,y,\zeta_{ab,A}), \end{aligned} \tag{4}$$

and the explicit fourth-order Milne formula with the error term:

$$
\begin{aligned}
g(x,y,z_0+h) = \quad & g(x,y,z_0-3h) \\
&+h\left(\frac{8}{3}g'(x,y,z_0) - \frac{4}{3}g'(x,y,z_0-h) + \frac{8}{3}g'(x,y,z_0-2h)\right. \\
&\left.+0g'(x,y,z_0-3h)\right) + \frac{224h^5}{720}g^{(5)}(x,y,\zeta_{m,4}).
\end{aligned}
\tag{5}
$$

where $\frac{251h^5}{720}g^{(5)}(x,y,\zeta_{ab,4})$ and $\frac{224h^5}{720}g^{(5)}(x,y,\zeta_{m,4})$ are truncation errors, respectively, and $\zeta_{ab,4} \in [z_0-3h,\ z_0+h]$ and $\zeta_{m,4} \in [z_0-3h,\ z_0+h]$ represent constant numbers, respectively. The positive constant $h$ is the step length (step size) for math formulae and is the depth for downward continuation. For the use of downward continuation, error terms in (4) and (5) are omitted, and the remaining parts of (4) and (5) are called the explicit fourth-order Adams–Bashforth method and the explicit fourth-order Milne method, respectively.

A method of order $k$ is that this method has a truncation error of $O(h^{k+1})$ but the number of the method's steps represented by $s$ denotes the number of known points. The number of steps $s$ of the explicit Adams–Bashforth (4) and that of the explicit Milne (5) is 4, which is equal to the orders $k$ of the above expressions. It should be noted that the equality of $k$ and $s$ for other expressions of multistep methods is not always true.

*2.2. Two Implicit Expressions and Their Predictor-Corrector Methods for Downward Continuation*
2.2.1. Two Implicit Expressions for Gravity Anomalies

The explicit expressions (4) and (5) represent polynomial extrapolations that are used to estimate downward continuations by integrating. However, it is well known that extrapolations tend to have poor accuracy of approximation since they are polynomials that are inferred outside the interval of a given dataset [17]. Therefore, it is recommended to investigate alternative methods that can estimate the values of unknown points inside the dataset and provide better approximations. These alternative methods are called polynomial interpolations.

In Equation (3) of multistep methods of numerical solutions, if $\beta_{-1} \neq 0$, methods used to estimate gravity anomaly downward continuations are interpolations. These interpolation methods represent implicit expressions for downward continuation.

To obtain the implicit expressions, recall the identity Equation (1) of the mean-value theorem for gravity anomalies. We can calculate the value of $g(x,y,z_0+h)$ by replacing the integrated form of the second term of Equation (1) based on the fundamental theorem of calculus:

$$
g(x,y,z_0+h) = g(x,y,z_0) + \int_{z_0}^{z_0+h} g'(x,y,z)dz.
\tag{6}
$$

To evaluate the integral for an implicit expression, we use Newton's backward difference polynomial [22] to determine a cubic polynomial that interpolates three previous-step values and one forward-step value $g'(x,y,z)$ to points $(z_0)$, $(z_0-h)$, $(z_0-2h)$, and $(z_0+h)$, where the step $s=3$. Denoting corresponding values at these steps by $g'(x,y,z_0)$, $g'(x,y,z_0-h)$, $g'(x,y,z_0-2h)$, and $g'(x,y,z_0+h)$, the required cubic interpolating polynomial, $G_c'(x,y,z)$, approximating $g'(x,y,z)$ from Newton's backward difference formula, is:

$$
\begin{aligned}
G_c'(x,y,z) = \quad & \nabla^0 g'(x,y,z_0+h) + \frac{(z-z_0-h)}{h}\nabla^1 g'(x,y,z_0+h) \\
&+\frac{(z-z_0-h)(z-z_0)}{2!h^2}\nabla^2 g'(x,y,z_0+h) \\
&+\frac{(z-z_0-h)(z-z_0)(z-z_0+h)}{3!h^3}\nabla^3 g'(x,y,z_0+h),
\end{aligned}
\tag{7}
$$

where $\nabla$ is the backward difference operator. We can evaluate the backward difference of $g'(x,y,z_0+h)$ as follows:

$$\nabla^0 g'(x, y, z_0 + h) = g'(x, y, z_0 + h),$$

$$\nabla^1 g'(x, y, z_0 + h) = \nabla^0 g'(x, y, z_0 + h) - \nabla^0 g'(x, y, z_0)$$
$$= g'(x, y, z_0 + h) - g'(x, y, z_0),$$

$$\nabla^2 g'(x, y, z_0 + h) = \nabla^1 g'(x, y, z_0 + h) - \nabla^1 g'(x, y, z_0)$$
$$= g'(x, y, z_0 + h) - 2g'(x, y, z_0) + g'(x, y, z_0 - h),$$

$$\nabla^3 g'(x, y, z_0 + h) = \nabla^2 g'(x, y, z_0 + h) - \nabla^2 g'(x, y, z_0)$$
$$= g'(x, y, z_0 + h) - 3g'(x, y, z_0) + 3g'(x, y, z_0 - h)$$
$$- g'(x, y, z_0 - 2h).$$

$$(8)$$

Substituting (8) into (7), we obtain:

$$
\begin{aligned}
G_c'(x, y, z) =\ & g'(x,\ y, z_0 + h) + \frac{(z - z_0 - h)}{h}\left(g'(x, y, z_0 + h) - g'(x, y, z_0)\right) \\
& + \frac{(z - z_0 - h)(z - z_0)}{2h^2}\left(g'(x, y, z_0 + h) - 2g'(x, y, z_0)\right. \\
& \left. + g'(x, y, z_0 - h)\right) \\
& + \frac{(z - z_0 - h)(z - z_0)(z - z_0 + h)}{6h^3}\left(g'(x, y, z_0 + h)\right. \\
& \left. - 3g'(x, y, z_0) + 3g'(x, y, z_0 - h) - g'(x, y, z_0 - 2h)\right).
\end{aligned}
$$

$$(9)$$

We now substitute this cubic polynomial approximation $G_c'(x, y, z)$ into $g'(x, y, z)$ in the integral Equation (6) and obtain:

$$
\begin{aligned}
rlg(x, y, z_0 + h) =\ & g(x, y, z_0) + \int_{z_0}^{z_0 + h} G_c'(x, y, z)dz \\
=\ & g(x, y, z_0) + g'(x, y, z_0 + h)CH_0 \\
& + \left(g'(x, y, z_0 + h) - g'(x, y, z_0)\right)CH_1 \\
& + \left(g'(x, y, z_0 + h) - 2g'(x, y, z_0) + g'(x, y, z_0 - h)\right)CH_2 \\
& + \left(g'(x, y, z_0 + h) - 3g'(x, y, z_0) + 3g'(x, y, z_0 - h)\right. \\
& \left. - g'(x, y, z_0 - 2h)\right)CH_3,
\end{aligned}
$$

$$(10)$$

where:

$$cCH_0 = \int_{z_0}^{z_0 + h} dz = h,$$

$$CH_1 = \int_{z_0}^{z_0 + h} \frac{(z - z_0 - h)}{h}dz = -\frac{h}{2},$$

$$CH_2 = \int_{z_0}^{z_0 + h} \frac{(z - z_0 - h)(z - z_0)}{2h^2}dz = -\frac{h}{12},$$

$$CH_3 = \int_{z_0}^{z_0 + h} \frac{(z - z_0 - h)(z - z_0)(z - z_0 + h)}{6h^3}dz = -\frac{h}{24}.$$

Substituting these integral evaluations into expression (10), we obtain:

$$
\begin{aligned}
g(x, y, z_0 + h) =\ & g(x, y, z_0) \\
& + h\left(\frac{9}{24}g'(x, y, z_0 + h) + \frac{19}{24}g'(x, y, z_0) - \frac{5}{24}g'(x, y, z_0 - h)\right. \\
& \left. + \frac{1}{24}g'(x, y, z_0 - 2h)\right).
\end{aligned}
$$

$$(11)$$

This is the implicit fourth-order expression of the Adams–Moulton method.

To determine its truncation error, $\varepsilon_{Gp}$, and order, $k$, we note that the error in the polynomial approximation $G_c'(x, y, z)$ at each point $z \in [z_0 - 2h, \ z_0 + h]$ can be determined from:

$$E(x, y, z) = \frac{g'^{(4)}\left(x, y, \zeta_{f,4}\right)}{4!}(z - z_0 - h)(z - z_0)(z - z_0 + h)(z - z_0 + 2h), \quad (12)$$

where $\zeta_{f,4} \in [z_0 - 2h, \ z_0 + h]$ represents the error in Newton's backward difference formula and can be used to bind the error for either interpolation or extrapolation depending on the choice of interval $[z_0 - 2h, \ z_0 + h]$. This error $E(x, y, z)$ can be incorporated into the integral of $G_c'(x, y, z)$ in (10) to obtain:

$$c\varepsilon_{Gp} = \int_{z_0}^{z_0+h} \frac{g'^{(4)}(x, y, \zeta_{f,4})}{4!}(z - z_0 - h)(z - z_0)(z - z_0 + h)(z - z_0 \\ + 2h)dz, \quad (13)$$

where $\zeta_{f,4}$ is a function of $z$. Denoting the minimum and maximum values of $\frac{g^{(4)}(x,y,\zeta_{f,4})}{4!}$ in the interval $[z_0 - 2h, \ z_0 + h]$ by $m$ and $M$, respectively, we obtain:

$$m \int_{z_0}^{z_0+h} p(x, y, z)dz \leq \varepsilon_{Gp} \leq M \int_{z_0}^{z_0+h} p(x, y, z)dz, \quad (14)$$

where $p(x, y, z) = (z - z_0 - h)(z - z_0)(z - z_0 + h)(z - z_0 + 2h)$. Since we assume $\frac{g^{(4)}(x,y,\zeta_{f,4})}{4!}$ to be continuous, taking all values between $m$ and $M$, then there must be a value of $z \in [z_0 - 2h, \ z_0 + h]$, say $\zeta_{am,4}$, for which:

$$rl\varepsilon_{Gp} = \int_{z_0}^{z_0+h} \frac{\frac{\partial g^{(4)}}{\partial z}\left(x, y, \zeta_{f,4}\right)}{4!} p(x, y, z)dz \\ = \frac{g^{(5)}(x, y, \zeta_{am,4})}{4!} \int_{z_0}^{z_0+h} p(x, y, z)dz. \quad (15)$$

Evaluating the definite integral of $p(x, y, z)$ above can be carried out by symbolic calculus computation to obtain:

$$\int_{z_0}^{z_0+h} p(x, y, z)dz = -\frac{19h^5}{30}, \quad (16)$$

hence,

$$\varepsilon_{Gp} = -\frac{19h^5}{720} g^{(5)}(x, y, \zeta_{am,4}), \ \zeta_{am,4} \in [z_0 - 2h, \ z_0 + h], \quad (17)$$

representing the local truncation error for the Adams–Moulton method and the order $k = 4$. So:

$$g(x, y, z_0 + h) = \ g(x, y, z_0) + h\left(\frac{9}{24}g'(x, y, z_0 + h) + \frac{19}{24}g'(x, y, z_0)\right. \\ \left. -\frac{5}{24}g'(x, y, z_0 - h) + \frac{1}{24}g'(x, y, z_0 - 2h)\right) \\ -\frac{19h^5}{720}g^{(5)}(x, y, \zeta_{am,4}). \quad (18)$$

This is the implicit fourth-order Adams–Moulton formula with the error term.

For another implicit expression, the Simpson method, we recall the integral Equation (6) and rewrite it as:

$$g(x, y, z_0 + h) = g(x, y, z_0 - h) + \int_{z_0-h}^{z_0+h} g'(x, y, z)dz. \tag{19}$$

We apply Simpson's rule [22,23] to the above integral. Simpson's rule is produced by a second Lagrange polynomial $G'_s(x, y, z)$ to approximate $g'(x, y, z)$ on $[z_0 - h, \ z_0 + h]$ with equally spaced intervals at three points, $(z_0 - h)$, $(z_0)$, and $(z_0 + h)$, where the step $s = 2$, denoting corresponding values by $g'(x, y, z_0 - h)$, $g'(x, y, z_0)$, and $g'(x, y, z_0 + h)$.

$$G'_s(x, y, z) = \sum_{i=z_0-h}^{z_0+h} g'(x, y, i) L_i(x, y, z), \tag{20}$$

where $L_i(x, y, z) = \prod_{k=z_0-h, \ k \neq i}^{z_0+h} \frac{z-k}{i-k}$ is Lagrange's interpolation formula (Lagrange polynomials). We can evaluate it as follows:

$$
\begin{aligned}
L_{z_0-h}(x, y, z) &= \frac{(z-z_0)(z-(z_0+h))}{((z_0-h)-z_0)((z_0-h)-(z_0+h))} \\
&= \frac{z^2 - (2z_0+h)z + z_0(z_0+h)}{2h^2} \\
L_{z_0}(x, y, z) &= \frac{(z-(z_0-h))(z-(z_0+h))}{(z_0-(z_0-h))(z_0-(z_0+h))} = \frac{z^2 - 2z_0z + (z_0^2 - h^2)}{-h^2}, \\
L_{z_0+h}(x, y, z) &= \frac{(z-(z_0-h))(z-z_0)}{((z_0+h)-(z_0-h))((z_0+h)-z_0)} \\
&= \frac{z^2 - (2z_0+h)z + z_0(z_0+h)}{2h^2}.
\end{aligned}
\tag{21}
$$

Therefore, the integral Equation (19) can be:

$$
\begin{aligned}
rlg(x, y, z_0 + h) &= g(x, y, z_0 - h) + \int_{z_0-h}^{z_0+h} G'_s(x, y, z)dz \\
&= g(x, y, z_0 - h) + \sum_{i=z_0-h}^{z_0+h} g'(x, y, i) \int_{z_0-h}^{z_0+h} L_i(x, y, z)dz \\
&= g(x, y, z_0 - h) + g'(x, y, z_0 - h)SH_0 + g'(x, y, z_0)SH_1 \\
&\quad + g'(x, y, z_0 + h)SH_2,
\end{aligned}
\tag{22}
$$

where:

$$cSH_0 = \int_{z_0-h}^{z_0+h} \frac{z^2 - (2z_0+h)z + z_0(z_0+h)}{2h^2} dz = \frac{h}{3},$$

$$SH_1 = \int_{z_0-h}^{z_0+h} \frac{z^2 - 2z_0z + (z_0^2 - h^2)}{-h^2} dz = \frac{4h}{3},$$

$$SH_2 = \int_{z_0-h}^{z_0+h} \frac{z^2 - (2z_0+h)z + z_0(z_0+h)}{2h^2} dz = \frac{h}{3}.$$

Substituting these into the expression (22), we obtain:

$$
\begin{aligned}
g(x, y, z_0 + h) = \ & g(x, y, z_0 - h) \\
&+ h\left(\frac{1}{3}g'(x, y, z_0 + h) + \frac{4}{3}g'(x, y, z_0) + \frac{1}{3}g'(x, y, z_0 - h)\right. \\
&+ 0\bigg).
\end{aligned}
\tag{23}
$$

This is the implicit fourth-order expression of the Simpson method.

For its truncation error, we similarly use the interpolation into the integral in (19) to obtain:

$$\varepsilon_{Gq} = \int_{z_0-h}^{z_0+h} \frac{g'^{(3)}(x,y,\zeta_{s,4})}{3!}(z-z_0+h)(z-z_0)(z-z_0-h)dz, \tag{24}$$

where $\zeta_{s,4}$ is a function of $z$. We also assume the minimum and maximum values of $\frac{g^{(3)}(x,y,\zeta_{s,4})}{3!}$ in the interval $[z_0-h,\ z_0+h]$ by $m$ and $M$, respectively:

$$m\int_{z_0-h}^{z_0+h} q(x,y,z)dz \le \varepsilon_{Gq} \le M\int_{z_0-h}^{z_0+h} q(x,y,z)dz, \tag{25}$$

where $q(x,y,z) = (z-z_0+h)(z-z_0)(z-z_0-h)$. Similarly to the Adams–Moulton method, we evaluate the definite integral of $q(x,y,z)$ but we obtain zero for $\int_{z_0-h}^{z_0+h} q(x,y,z)dz = 0$. So we instead define:

$$w(x,y,z) = \int_{z_0-h}^{z} q(x,y,z)dz. \tag{26}$$

Integrating (24) by parts we obtain:

$$\begin{aligned}
ll\varepsilon_{Gq} &= \int_{z_0-h}^{z_0+h} \frac{g'^{(3)}(x,y,\zeta_{s,4})}{3!}w'(x,y,z)dz \\
&= w(x,y,z)\frac{g'^{(3)}(x,y,\zeta_{s,4})}{3!}\Bigg|_{z=z_0-h}^{z=z_0+h} \\
&\quad - \int_{z_0-h}^{z_0+h} w(x,y,z)\frac{\partial}{\partial z}\frac{g'^{(3)}(x,y,\zeta_{s,4})}{4!}dz \\
&= -\int_{z_0-h}^{z_0+h} w(x,y,z)\frac{\partial}{\partial z}\frac{\frac{\partial g}{\partial z}^{(3)}(x,y,\zeta_{s,4})}{4!}dz.
\end{aligned} \tag{27}$$

Applying the integral mean-value theorem we obtain:

$$\varepsilon_{Gq} = -\frac{g^{(5)}(x,y,\zeta_{s,4})}{4!}\int_{z_0-h}^{z_0+h} w(x,y,z)dz = -\frac{g^{(5)}(x,y,\zeta_{s,4})}{24}\frac{4h^5}{15}, \tag{28}$$

hence,

$$\varepsilon_{Gq} = -\frac{h^5}{90}g^{(5)}(x,y,\zeta_{s,4}),\ \zeta_{s,4} \in [z_0-h,\ z_0+h], \tag{29}$$

represents the local truncation error for the Simpson method and the order $k=4$, so:

$$\begin{aligned}
g(x,y,z_0+h) &= g(x,y,z_0-h) \\
&\quad + h\left(\frac{1}{3}g'(x,y,z_0+h) + \frac{4}{3}g'(x,y,z_0) + \frac{1}{3}g'(x,y,z_0-h) + 0\right) \\
&\quad - \frac{h^5}{90}g^{(5)}(x,y,\zeta_{s,4}).
\end{aligned} \tag{30}$$

This is the implicit fourth-order Adams–Moulton formula with the error term.

The remaining parts of (18) and (30), whose error terms are omitted, represented by (11) and (23), are called the implicit fourth-order Adams–Moulton method and the implicit fourth-order Simpson method, respectively, for downward continuation.

It can be seen the implicit fourth-order Adams–Moulton method (11) and the implicit fourth-order Simpson method (23) have local truncation errors of $-\frac{19h^5}{720}g^{(5)}(x,y,\zeta_{am,4})$ and $-\frac{h^5}{90}g^{(5)}(x,y,\zeta_{s,4})$, respectively, which are smaller than the corresponding ones of $\frac{251h^5}{720}g^{(5)}(x,y,\zeta_{ab,4})$ and $\frac{224h^5}{720}g^{(5)}(x,y,\zeta_{m,4})$ from the explicit fourth-order Adams–Bashforth method of (4) and the explicit fourth-order Milne method of (5). However, the implicit

fourth-order expressions of the Adams–Moulton (11) and the Simpson (23) methods contain vertical derivatives of gravity anomalies of being downward continued at the height of $z_0 + h$, which is the position of downward continuation depth, at which gravity anomalies and their vertical derivatives are unknown. Thus, they cannot be used directly for gravity anomaly downward continuation.

### 2.2.2. Predictor-Corrector Methods for Downward Continuation

To solve the implicit expression, which is hard to directly operate, we use results $g(x, y, z_0 + h)$ from explicit expressions of (11) and (23) to calculate vertical derivatives at the height of $z_0 + h$, which is the position of downward continuation depth by the ISVD method [18]. Other methods for calculating vertical derivatives can also be used. Taking calculated vertical derivatives $g'_*(x, y, z_0 + h)$ as input of implicit expressions, we combine explicit and implicit expressions as 'predictor-corrector' pairs. The Adams–Bashforth predictor and the Adams–Moulton corrector are, respectively, written as:

$$
\begin{aligned}
g_{Pab}(x, y, z_0 + h) \quad &= g(x, y, z_0) + h\left(\frac{55}{24}g'(x, y, z_0) - \frac{59}{24}g'(x, y, z_0 - h)\right. \\
&\left. + \frac{37}{24}g'(x, y, z_0 - 2h) - \frac{9}{24}g'(x, y, z_0 - 3h)\right) \\
&+ \frac{251h^5}{720}g^{(5)}(x, y, \zeta_{ab,4})
\end{aligned}
\tag{31}
$$

$$
\begin{aligned}
g_{Cam}(x, y, z_0 + h) \\
&= g(x, y, z_0) \\
&+ h\left(\frac{9}{24}g'_{*,ab}(x, y, z_0 + h) + \frac{19}{24}g'(x, y, z_0)\right. \\
&\left. - \frac{5}{24}g'(x, y, z_0 - h) + \frac{1}{24}g'(x, y, z_0 - 2h)\right) \\
&- \frac{19h^5}{720}g^{(5)}(x, y, \zeta_{am,4}).
\end{aligned}
\tag{32}
$$

where $g_{Pab}(x, y, z_0 + h)$ and $g_{Cam}(x, y, z_0 + h)$ are the Adams–Bashforth predictor and Adams–Moulton corrector of the gravity anomaly to be downward continued at the height of $z_0 + h$. $g_{Cam}(x, y, z_0 + h)$ in the expression (32) is the final result of downward continuation by the Adams–Bashforth–Moulton predictor-corrector method. Note that the $g'_{*,ab}(x, y, z_0 + h)$ term in the corrector expression (32) is estimated by calculating vertical derivatives as $\left.\frac{\partial g_{Pab}(x,y,z)}{\partial z}\right|_{z=z_0+h}$ by the ISVD method from the predictor value $g_{Pab}(x, y, z_0 + h)$ in the expression (31).

Another pair of the predictor-corrector, the Milne predictor and the Simpson corrector are:

$$
\begin{aligned}
g_{Pm}(x, y, z_0 + h) \quad &= g(x, y, z_0 - 3h) \\
&+ h\left(\frac{8}{3}g'(x, y, z_0) - \frac{4}{3}g'(x, y, z_0 - h) + \frac{8}{3}g'(x, y, z_0 - 2h)\right. \\
&\left. + 0\right) + \frac{224h^5}{720}g^{(5)}(x, y, \zeta_{m,4}),
\end{aligned}
\tag{33}
$$

$$
\begin{aligned}
g_{Cs}(x, y, z_0 + h) \quad &= g(x, y, z_0 - h) \\
&+ h\left(\frac{1}{3}g'_{*,m}(x, y, z_0 + h) + \frac{4}{3}g'(x, y, z_0) + \frac{1}{3}g'(x, y, z_0 - h)\right. \\
&\left. + 0\right) - \frac{8h^5}{720}g^{(5)}(x, y, \zeta_{s,4}),
\end{aligned}
\tag{34}
$$

where $g_{Pm}(x, y, z_0 + h)$ and $g_{Cs}(x, y, z_0 + h)$ are the Milne predictor and the Simpson corrector of the gravity anomaly to be downward continued at the height of $z_0 + h$. $g_{Cs}(x, y, z_0 + h)$ in the expression (34) is the final result of downward continuation by the Milne–Simpson predictor-corrector method. Note that the $g'_{*,m}(x, y, z_0 + h)$ term in corrector expression (32) is estimated by calculating vertical derivatives as $\left. \frac{\partial g_{Pab}(x,y,z)}{\partial z} \right|_{z=z_0+h}$ by the ISVD method from the predictor value $g_{Pm}(x, y, z_0 + h)$ in the expression (33).

The local truncation errors $g^{(5)}(x, y, \zeta_{f,4})$ of these two predictor-corrector pairs are different. The local truncation errors involved with a predictor-corrector pair of the Milne–Simpson method are generally smaller than those of the Adams–Bashforth–Moulton method. However, mathematically, the technique of Milne–Simpson has limited use because of round-off error problems, which do not occur with the Adams–Bashforth–Moulton method [24]. Thus, comparisons of gravity anomaly downward continuations between these methods should be carried out.

### 3. Examples and Comparison

To verify the validity of the predictor-corrector methods of the Milne–Simpson and the Adams–Bashforth–Moulton methods presented in this study and to compare them with the previous Milne and Adams–Bashforth methods, we test them on synthetic models and real data.

### 3.1. Synthetic Models

The synthetic model (Figure 1) consists of three cuboids with different sizes, different buried depths of top interfaces, and different density contrasts with the surrounding rock. The axis of z is upward throughout the synthetic model section. The measurement height of the ground observation surface equals 0 m (z equals 0 m). The yellow cuboid of the model has side lengths of 40 m, 10 m, and 20 m in the x, y, and z directions, respectively, with a center point at $(75, 65, -25)$ m. Its top interface is 15 m beneath the ground observation surface, and its density contrast with the surrounding rock is 0.6 g/cm$^3$. The green cuboid's side lengths in the x, y, and z directions are 10 m, 20 m, and 20 m, respectively, with a center point at $(65, 90, -22)$ m. Its top interface is 12 m beneath the ground observation surface, and its density contrast is 0.5 g/cm$^3$. The x, y, and z direction side lengths of the blue cuboid are 10 m, 12 m, and 20 m, respectively, and the center point coordinate of this cuboid is $(85, 90, -21)$ m. This cuboid's buried depth of its top interface is 11 m, and its density contrast is 0.4 g/cm$^3$. The observation surface has 150 measurement lines, 150 points per line, and a grid spacing of 1 m.

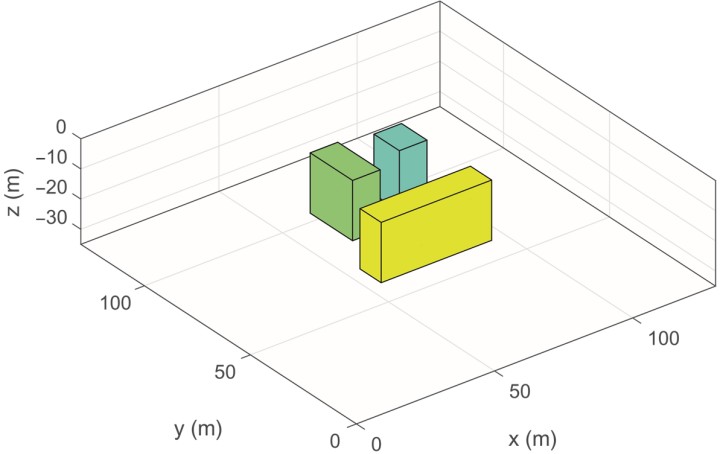

**Figure 1.** Subsurface distribution of the synthetic cuboid model.

In model examples, to compare different gravity anomaly downward continuations, we use six methods, including the classical FFT method [10], the integral iterative method [13],

the fourth-order Milne method [15], the fourth-order Milne–Simpson predictor-corrector method, the fourth-order Adams–Bashforth method [14,17], and the fourth-order Adams–Bashforth–Moulton predictor-corrector method. All theoretical gravity anomalies and vertical derivatives are forward calculated from the three-cuboid model by forward formulae in [25] in the synthetic model part. The unit depth of downward continuation is called a depth interval which is equal to the horizontal surface grid spacing of 1 m.

As the inputs of the four methods are based on numerical solutions of the mean-value theorem, the fourth-order Milne method, the fourth-order Milne–Simpson predictor-corrector method, the fourth-order Adams–Bashforth method, and the fourth-order Adams–Bashforth–Moulton predictor-corrector method are gravity anomalies and, with their vertical derivatives at different heights, different approaches of obtaining these inputs may have different downward continuation results because of calculation factors. To further analyze and understand the effect of introduced calculation factors, we also compare these methods under different conditions, such as with purely theoretical gravity anomalies and their vertical derivatives at different heights from forward calculations; the theoretical gravity anomaly and its vertical derivative at the measurement height of 0 m from forward calculations and corresponding gravity anomalies and their vertical derivatives at non-measurement heights above 0 m calculated by upward continuation; the theoretical gravity anomaly at the measurement height of 0 m from the forward calculation and corresponding calculated vertical derivatives of the gravity anomaly at the measurement height of 0 m by the ISVD method and corresponding gravity anomalies and their vertical derivatives at non-measurement heights above 0 m calculated by upward continuation; adding noise to the third condition.

### 3.1.1. Downward Continuation with Theoretical Gravity Anomalies and Their Vertical Derivatives at Different Heights from Forward Calculations

For the classical FFT method, no regularization or filters are added. Furthermore, no regularization or filters are added for the following methods. For the integral iteration method, the step length of iteration is set to 1, the convergence error is 0.0001 mGal, and its convergent iteration stops at 10 times from a total set of 50 times. For the four methods with the same fourth-order based on numerical solutions including the previous two, the Milne method and the Adams–Bashforth method, and another two new proposed ones, the Milne–Simpson predictor-corrector method and the Adams–Bashforth–Moulton predictor-corrector method, the input values for downward continuation are theoretical values, which means that gravity anomalies and their vertical derivatives at different heights except at the downward depth are all forward calculated.

We use the above six methods to downward continue the theoretical gravity anomaly (Figure 2a) from the ground observation surface, whose measurement height equals 0 m, to the depth of 8 m which is 8 m beneath the ground observation surface. The downward continuation results obtained by the classical FFT method, the integral iteration method, the Milne method, the Milne–Simpson predictor-corrector method, the Adams–Bashforth method, and the Adams–Bashforth–Moulton predictor-corrector method are shown in Figure 2c–h. To verify these results, Figure 2b represents the theoretical downward continuation values at the height of −8 m, called the reference gravity anomaly herein. The RMS (root-mean-square) errors between the reference gravity anomaly (Figure 2b) and six downward continuation results (Figure 2c–h) are respectively shown in the second column (Section 3.1.1 column) of Table 1.

Compared with the reference gravity anomaly (Figure 2b), the classical FFT method's downward continuation (Figure 2c) is divergent. The downward continuation (Figure 2d) of the integral iteration method is accurate, but there are some distortions in the edge areas. For the Milne method (Figure 2e) and the Adams–Bashforth method (Figure 2g), their downward continuations are relatively accurate and stable, but gravity anomalies caused by the small-scale source of the green cuboid are not indicated well. Downward continuations by the Milne–Simpson predictor-corrector method (Figure 2f) and the Adams–

Bashforth–Moulton predictor-corrector method (Figure 2h) are accurate and stable and provide the best results compared with the reference gravity anomaly (Figure 2b).

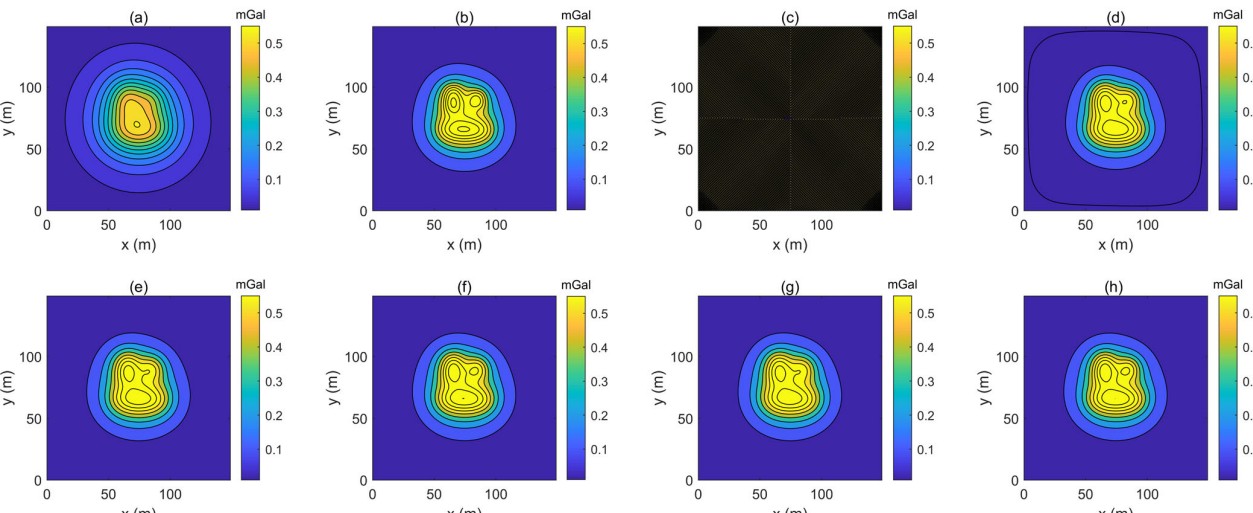

**Figure 2.** Theoretical gravity anomalies and their vertical derivatives from forward calculations at different heights are used in downward continuations. All downward continuation depths are 8 m (eight times the depth interval) at the height of −8 m. (**a**) The gravity anomaly to be downward continued, which is the theoretical gravity anomaly from forward calculation at the measurement height of 0 m regarded as the ground observation surface, (**b**) The reference gravity anomaly for downward continuations, which is the theoretical gravity anomaly from forward calculation at the height of −8 m. (**c**) The downward continuation of (**a**) by the classical FFT method. (**d**) The downward continuation of (**a**) by the integral iteration method. (**e**) The downward continuation of (**a**) by the Milne method. (**f**) The downward continuation of (**a**) by the Milne–Simpson predictor-corrector method. (**g**) The downward continuation of (**a**) by the Adams–Bashforth method. (**h**) The downward continuation of (**a**) by the Adams–Bashforth–Moulton predictor-corrector method.

**Table 1.** The RMS errors between the reference gravity anomaly and six downward continuation results at the height of −8 m under different conditions: Section 3.1.1 theoretical gravity anomalies and their vertical derivatives at different heights from forward calculations; Section 3.1.2 the theoretical gravity anomaly and its vertical derivative at the measurement height of 0 m from forward calculations and corresponding gravity anomalies and their vertical derivatives at non-measurement heights above 0 m calculated by upward continuation; Section 3.1.3 the theoretical gravity anomaly at the measurement height of 0 m from the forward calculation and corresponding calculated vertical derivatives of the gravity anomaly at the measurement height of 0 m by the ISVD method and corresponding gravity anomalies and their vertical derivatives at non-measurement heights above 0 m calculated by upward continuation; Section 3.1.4 adding noise to the third condition.

| RMS Errors / Methods | Section 3.1.1 | Section 3.1.2 | Section 3.1.3 | Section 3.1.4 |
|---|---|---|---|---|
| FFT | $0.42 \times 10^{17}$ | $0.42 \times 10^{17}$ | $0.42 \times 10^{17}$ | $0.19 \times 10^{20}$ |
| Integral iteration | $0.16 \times 10^{-2}$ | $0.16 \times 10^{-2}$ | $0.16 \times 10^{-2}$ | $0.17 \times 10^{-2}$ |
| Milne | $0.92 \times 10^{-3}$ | $0.39 \times 10^{-2}$ | $0.30 \times 10^{-2}$ | $0.30 \times 10^{-2}$ |
| Milne–Simpson predictor-corrector | $0.52 \times 10^{-3}$ | $0.13 \times 10^{-2}$ | $0.10 \times 10^{-2}$ | $0.16 \times 10^{-2}$ |
| Adams–Bashforth | $0.95 \times 10^{-3}$ | $0.95 \times 10^{-3}$ | $0.10 \times 10^{-2}$ | $0.11 \times 10^{-2}$ |
| Adams–Bashforth–Moulton predictor-corrector | $0.53 \times 10^{-3}$ | $0.53 \times 10^{-3}$ | $0.61 \times 10^{-3}$ | $0.13 \times 10^{-2}$ |

The RMS errors (the second column of Table 1) between the reference gravity anomaly (Figure 2b) and six downward continuation results (Figure 2c–h) show that the four methods based on numerical solutions of the mean-value theorem have better downward

continuation than the classic FFT method and the integral iteration method. For the four methods based on numerical solutions, two newly presented methods are better than the previous methods, and the Milne–Simpson predictor-corrector method provides the best downward continuation.

### 3.1.2. Downward Continuation with the Theoretical Gravity Anomaly and Its Vertical Derivative at the Measurement Height of 0 m from Forward Calculations

The condition of Section 3.1.1 is too ideal to realise as there are few gravity anomalies and their vertical derivatives are known at different heights of the same distance simultaneously over one area in real. To simulate real cases, we assume that the gravity anomaly and its vertical derivative only on the ground observation surface are known, which means the theoretical gravity anomaly (Figure 3a) and its vertical derivative at the measurement height of 0 m are from forward calculations, but the corresponding gravity anomalies and their vertical derivatives used in downward continuation at non-measurement heights above 0 m are calculated by upward continuation.

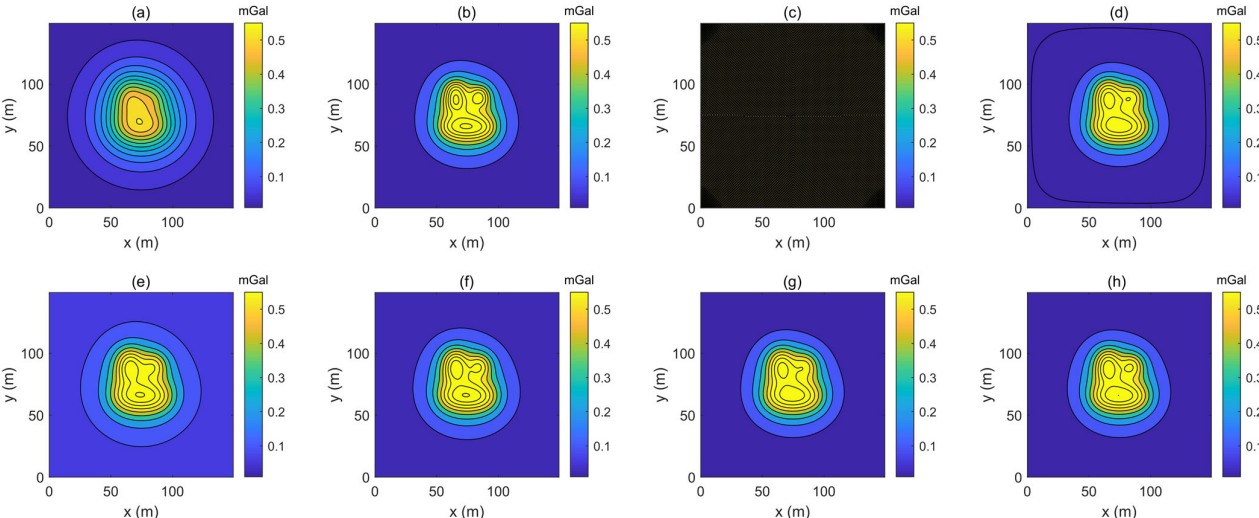

**Figure 3.** The theoretical gravity anomaly and its vertical derivative at the measurement height of 0 m from the forward calculation and corresponding gravity anomalies and their vertical derivatives at non-measurement heights above 0 m calculated by upward continuation are used in downward continuations. All downward continuation depths are 8 m (eight times the depth interval) which is at the measurement height of −8 m. (**a**) The gravity anomaly to be downward continued, which is the theoretical gravity anomaly from forward calculation at the measurement height of 0 m regarded as the ground observation surface. (**b**) The reference gravity anomaly for downward continuations, which is the theoretical gravity anomaly from forward calculation at the measurement height of −8 m. (**c**) The downward continuation of (**a**) by the classical FFT method. (**d**) The downward continuation of (**a**) by the integral iteration method. (**e**) The downward continuation of (**a**) by the Milne method. (**f**) The downward continuation of (**a**) by the Milne–Simpson predictor-corrector method. (**g**) The downward continuation of (**a**) by the Adams–Bashforth method. (**h**) The downward continuation of (**a**) by the Adams–Bashforth–Moulton predictor-corrector method.

The reference gravity anomaly at the height of −8 m in Figure 3b, the downward continuation by the classical FFT method in Figure 3c, and the downward continuation by the integral iteration method in Figure 3d are the same as those in Figure 2b–d, respectively. For downward continuations in Figure 3e–h by the Milne method, the Adams–Bashforth method, the Milne–Simpson predictor-corrector method, and the Adams–Bashforth–Moulton predictor-corrector method, parts of input values which are the theoretical gravity anomaly and its vertical derivative at the height of 0 m are from forward calculations but other inputs at different heights above 0 m are upward continued by the classical FFT method. The RMS (root-mean-square) errors between the reference gravity

anomaly (Figure 3b) and six downward continuation results (Figure 3c–h) are respectively shown in the third column (Section 3.1.2 column) of Table 1.

Compared with the reference gravity anomaly (Figure 3b), except the result (Figure 3c) from the classic FFT method, other downward continuations (Figure 3d–h) are accurate and stable. Among them, the newly presented Adams–Bashforth–Moulton predictor-corrector method provides the best result (Figure 3h).

The RMS errors in the third column of Table 1 show that, for the condition of the theoretical gravity anomaly and its vertical derivative at the measurement height of 0 m from forward calculations but corresponding gravity anomalies and their vertical derivatives used in downward continuation at non-measurement heights above 0 m calculated by upward continuation, the smallest RMS error is that of the Adams–Bashforth–Moulton predictor-corrector method, which provides the best downward continuation, and the following are the Adams–Bashforth method, the Milne–Simpson predictor-corrector method, the integral iteration method, the Milne method, and the classic FFT method.

The sorting order of RMS errors of four methods based on numerical solutions from small to large is 'the Adams-Bashforth-Moulton predictor-corrector method, the Adams-Bashforth method, the Milne-Simpson predictor-corrector method, and the Milne method', which is different from their corresponding truncation errors' behaviour increasing from $\frac{8h^5}{720}g^{(5)}(x,y,\zeta_{s,4})$ of the Milne–Simpson predictor-corrector method, $\frac{19h^5}{720}g^{(5)}(x,y,\zeta_{am,4})$ of the Adams–Bashforth–Moulton predictor-corrector method, $\frac{224h^5}{720}g^{(5)}(x,y,\zeta_{m,4})$ of the Milne method to $\frac{251h^5}{720}g^{(5)}(x,y,\zeta_{ab,4})$ of the Adams–Bashforth method in Section 3.1.1.

### 3.1.3. Downward Continuation with the Theoretical Gravity Anomaly at the Measurement Height of 0 m from the Forward Calculation

Vertical derivatives of gravity anomalies can not always be obtained from measurements in real cases. To simulate this general situation, we assume that only the theoretical gravity anomaly (Figure 4a) at the measurement height of 0 m is from the forward calculation. Used as input of four downward continuation methods based on numerical solutions of the mean-value theorem, the vertical derivative at the measurement height of 0 m is calculated by the ISVD method and corresponding gravity anomalies and their vertical derivatives at non-measurement heights above 0 m are calculated by the classical FFT method of upward continuation.

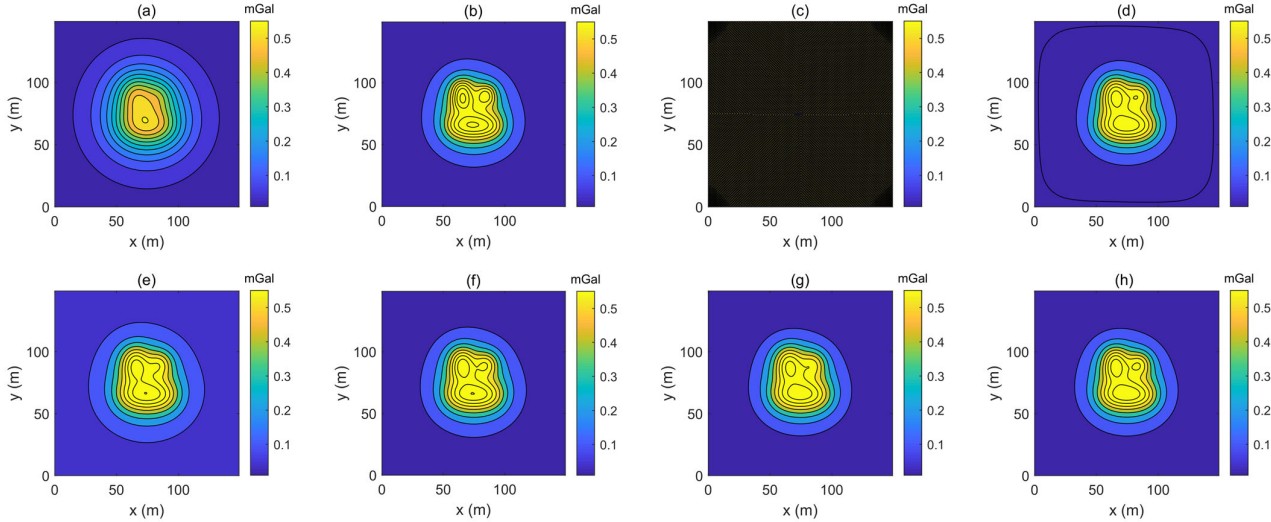

**Figure 4.** The theoretical gravity anomaly from forward calculation at the measurement height of 0 m, corresponding calculated vertical derivatives of the gravity anomaly at the measurement height of 0 m by the ISVD method, and corresponding gravity anomalies and their vertical derivatives at non-measurement heights above 0 m calculated by upward continuation are used in downward continuations.

All downward continuation depths are 8 m (eight times the depth interval) which is at the measurement height of −8 m. (**a**) The gravity anomaly to be downward continued, which is the theoretical gravity anomaly from forward calculation at the measurement height of 0 m regarded as the ground observation surface. (**b**) The reference gravity anomaly for downward continuations, which is the theoretical gravity anomaly from forward calculation at the measurement height of −8 m. (**c**) The downward continuation of (**a**) by the classical FFT method. (**d**) The downward continuation of (**a**) by the integral iteration method. (**e**) The downward continuation of (**a**) by the Milne method. (**f**) The downward continuation of (**a**) by the Milne–Simpson predictor-corrector method. (**g**) The downward continuation of (**a**) by the Adams–Bashforth method. (**h**) The downward continuation of (**a**) by the Adams–Bashforth–Moulton predictor-corrector method.

The six downward continuation results are shown in Figure 4c-h and the corresponding differences between six downward continuations and the reference gravity anomaly (4b) are shown in Figure 5. The RMS (root-mean-square) errors between the reference gravity anomaly (Figure 4b) and six downward continuation results (Figure 4c-h) are respectively shown in the fourth column (Section 3.1.3 column) of Table 1.

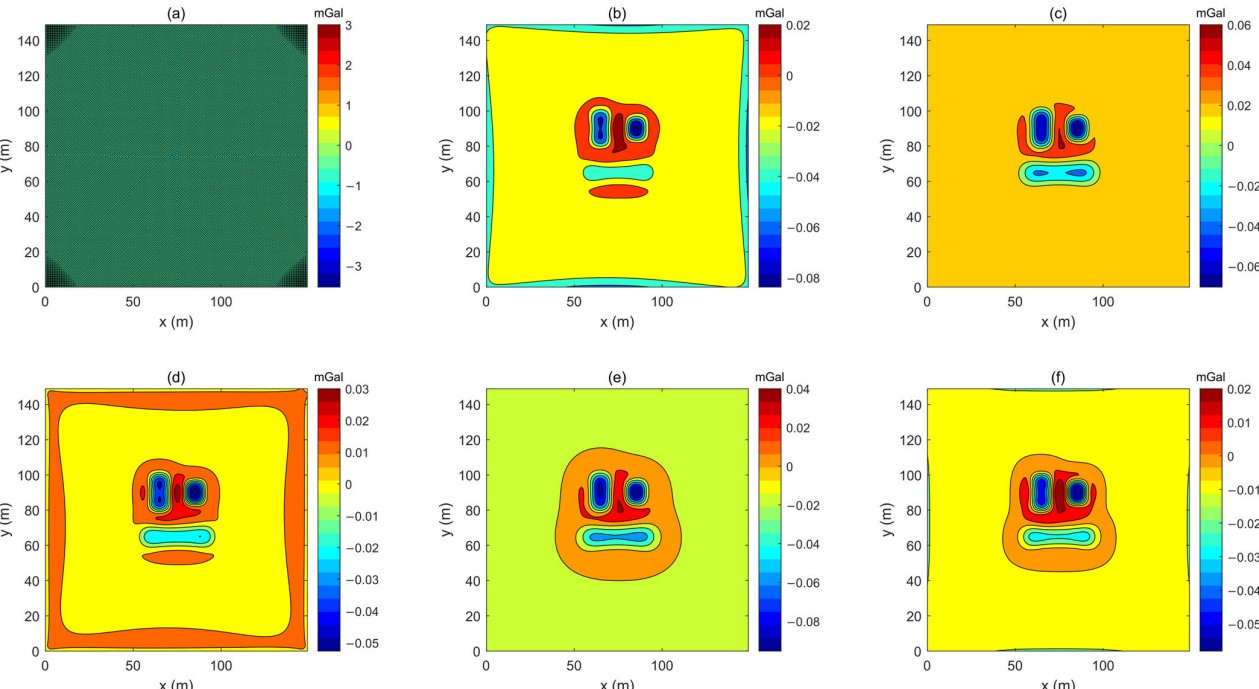

**Figure 5.** The differences between the downward continuations and the reference gravity anomaly from forward calculation at the height of −8 m. (**a**) The difference between Figure 4b,c. (**b**) The difference between Figure 4b,d. (**c**) The difference between Figure 4b,e. (**d**) The difference between Figure 4b,f. (**e**) The difference between Figure 4b,g. (**f**) The difference between Figure 4b,h.

The downward continuations (Figure 4c–h) and corresponding differences (Figure 5a–f) between the downward continuations and the reference gravity anomaly show that the two newly presented methods yield better downward continuations (Figure 4f,h) than the other methods, and the Adams–Bashforth–Moulton predictor-corrector method presents the best downward continuation (Figure 4h) among these six methods.

RMS errors in the fourth column of Table 1 show that four methods based on numerical solutions are accurate and stable, but the sorting of the accuracy for downward continuation decreases from the Adams–Bashforth–Moulton method and the Milne–Simpson method, equalling the Adams–Bashforth method to the Milne method. The Milne–Simpson method and the Adams–Bashforth method almost provide the same RMS errors, but the Adams–Bashforth–Moulton method still generates the smallest RMS error.

We also estimate consuming times for six methods on a PC with i7 CPU at 2.9 GHz and 32.00 GB RAM. The classical FFT method, the integral iteration method, the Milne method, the Milne–Simpson predictor-corrector method, the Adams–Bashforth method, and the Adams–Bashforth–Moulton predictor-corrector method are 0.0091 s, 0.0131 s, 0.0045 s, 1.2442 s, 0.0041 s, and 1.2532 s, respectively.

### 3.1.4. Downward Continuation with the Theoretical Gravity Anomaly at the Measurement Height of 0 m from the Forward Calculation with Gaussian White Noise

Though we would prefer not to downward continue gravity anomalies with noise, noise cannot be avoided or filtered completely. To simulate this condition, we assume that only the theoretical gravity anomaly at the measurement height of 0 m is from forward calculation but contaminated with 2% Gaussian white noise. The inputs of four downward continuation methods based on numerical solutions of the mean-value theorem are the vertical derivative at the measurement height of 0 m calculated by the ISVD method, and corresponding gravity anomalies and their vertical derivatives at non-measurement heights above 0 m calculated by the classical FFT method of upward continuation. Results of downward continuations are shown in Figure 6c-h, and the RMS (root-mean-square) errors between the reference gravity anomaly (Figure 6b) and six downward continuation results (Figure 6c-h) are respectively shown in the fifth column (Section 3.1.4 column) of Table 1.

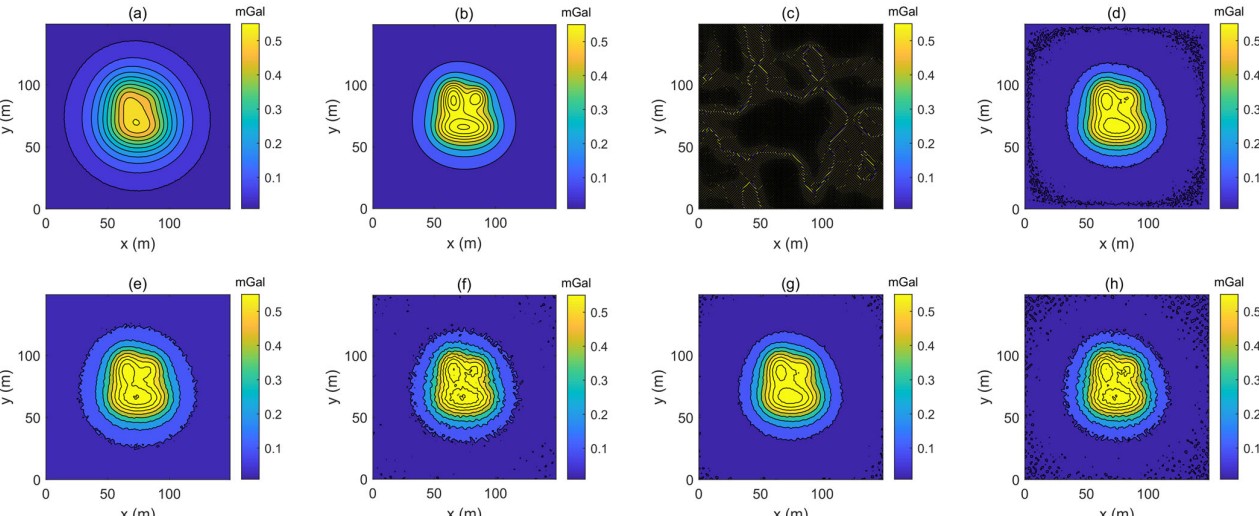

**Figure 6.** The theoretical gravity anomaly from forward calculation with 2% Gaussian white noise at the measurement height of 0 m, corresponding calculated vertical derivatives of the gravity anomaly at the measurement height of 0 m by the ISVD method, and corresponding gravity anomalies and their vertical derivatives at non-measurement heights above 0 m calculated by upward continuation are used in downward continuations. All downward continuation depths are 8 m (eight times the depth interval) which is at the measurement height of −8 m. (**a**) The gravity anomaly to be downward continued, which is the theoretical gravity anomaly from forward calculation with 2% Gaussian white noise at the measurement height of 0 m regarded as the ground observation surface. (**b**) The reference gravity anomaly for downward continuations, which is the theoretical gravity anomaly from forward calculation without noise at the measurement height of −8 m. (**c**) The downward continuation of (**a**) by the classical FFT method. (**d**) The downward continuation of (**a**) by the integral iteration method. (**e**) The downward continuation of (**a**) by the Milne method. (**f**) The downward continuation of (**a**) by the Milne–Simpson predictor-corrector method. (**g**) The downward continuation of (**a**) by the Adams–Bashforth method. (**h**) The downward continuation of (**a**) by the Adams–Bashforth–Moulton predictor-corrector method.

The results show that four methods (Figure 6e–h) based on numerical solutions can provide stable downward continuations. From RMS errors, the presented two methods

(Figure 6f,h) are better than the integral iteration method. The order of the accuracy for downward continuation decreases from the Adams–Bashforth method, the Adams–Bashforth–Moulton predictor-corrector method, the Milne–Simpson predictor-corrector method, to the Milne method. The Adams–Bashforth method is the best one (Figure 6g) with the smallest RMS error.

### 3.1.5. RMS Errors at Different Depths by Different Downward Continuation Methods

To better understand these downward continuation methods, especially the presented two methods and the previous two based on numerical solutions of the mean-value theorem, in Figure 7, we present variations of RMS errors of the integral iteration method, the Milne method, the Adams–Bashforth method, the Milne–Simpson predictor-corrector method, and the Adams–Bashforth–Moulton predictor-corrector method with the downward continuation depth from 1 m to 10 m on the conditions of Sections 3.1.1–3.1.3.

From Figure 7a,b, we can see that downward continuations based on four numerical solutions with purely theoretical gravity anomalies and their vertical derivatives at different heights from forward calculations are better than the integral iteration method. The RMS errors of the two presented methods of the Milne–Simpson predictor-corrector method and the Adams–Bashforth–Moulton are almost the same. Before the depth of 4 m, the two previous methods, the Milne method and the Adams–Bashforth method, have smaller RMS errors than the Milne–Simpson predictor-corrector method and the Adams–Bashforth–Moulton predictor-corrector method, but after the depth of 4 m, the two presented methods are the best among these methods.

From Figure 7c,d, we can see that with both the theoretical gravity anomaly and its vertical derivative at the height of 0 m available, the downward continuation based on the Adams–Bashforth method provides the smallest RMS errors before the depth of 4 m, but the presented Adams–Bashforth–Moulton predictor-corrector method gives the smallest RMS errors after the depth of 4 m.

From Figure 7e,f, when only the theoretical gravity anomaly at the height of 0 m is possible, the downward continuation by the presented Adams–Bashforth–Moulton predictor-corrector method has the smallest RMS error.

### 3.2. Real Data

To verify the actual downward continuation by the Milne–Simpson predictor-corrector method and the Adams–Bashforth–Moulton predictor-corrector method proposed in this study, we use the gravity anomaly from the airborne measurement at the height of 200 m over the Nechako Basin area of Canada. Both the airborne Bouguer gravity anomaly (Figure 8b) and its vertical derivative are available in this area. The grid spacings of the gravity anomaly and its vertical derivative are 400 m. For testing, we upward continue the gravity anomaly (Figure 8b) and its vertical derivative from their measurement height of 200 m to a height of 2200 m (equivalent to five spacing intervals) and consider this upward continuation gravity anomaly (Figure 8a) at the height of 2200 m as the one to be downward continued.

First, we use both the real gravity anomaly (Figure 8a) and the real vertical derivative, which are obtained by upward continuation at the height of 2200 m as inputs to downward continue by the Milne method, the Milne–Simpson predictor-corrector method, the Adams–Bashforth method, and the Adams–Bashforth–Moulton predictor-corrector method, respectively. Results are shown in Figure 8c–f. The two presented methods can provide stable and accurate downward continuations (Figure 8d,f).

To widely illustrate applications of the proposed methods, we carry out the downward continuations only using the observed gravity anomaly at the height of 2200 m. The corresponding results of the Milne method, the Milne–Simpson predictor-corrector method, the Adams–Bashforth method, and the Adams–Bashforth–Moulton predictor-corrector method are shown in Figure 9c–f. Additionally, reasonable results are obtained by the presented methods.

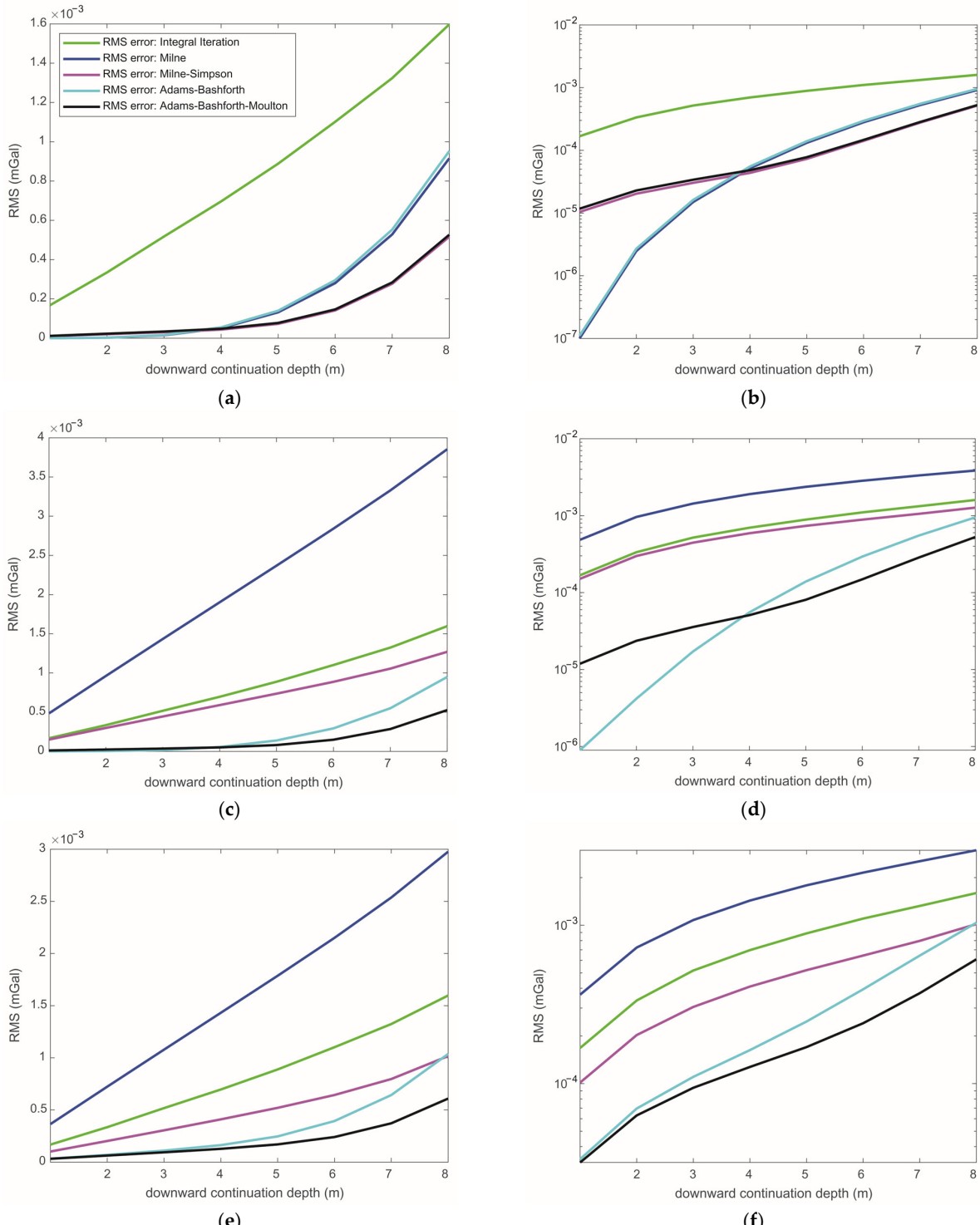

**Figure 7.** Variations of RMS errors between reference gravity anomalies from forward calculations and downward continuation results by different methods from the height of −1 m to that of −10 m. Green lines represent the integral iteration method, blue lines represent the Milne method, magenta lines represent the Milne–Simpson predictor-corrector method, cyan lines represent the Adams–Bashforth method, and black lines represent the Adams–Bashforth–Moulton predictor-corrector method. (**a**) Under the condition of Section 3.1.1 in general coordinates. (**b**) Under the condition of Section 3.1.1 in logarithmic coordinates. (**c**) Under the condition of Section 3.1.2 in general coordinates. (**d**) Under the condition of Section 3.1.2 in logarithmic coordinates. (**e**) Under the condition of Section 3.1.3 in general coordinates. (**f**) Under the condition of Section 3.1.3 in logarithmic coordinates.

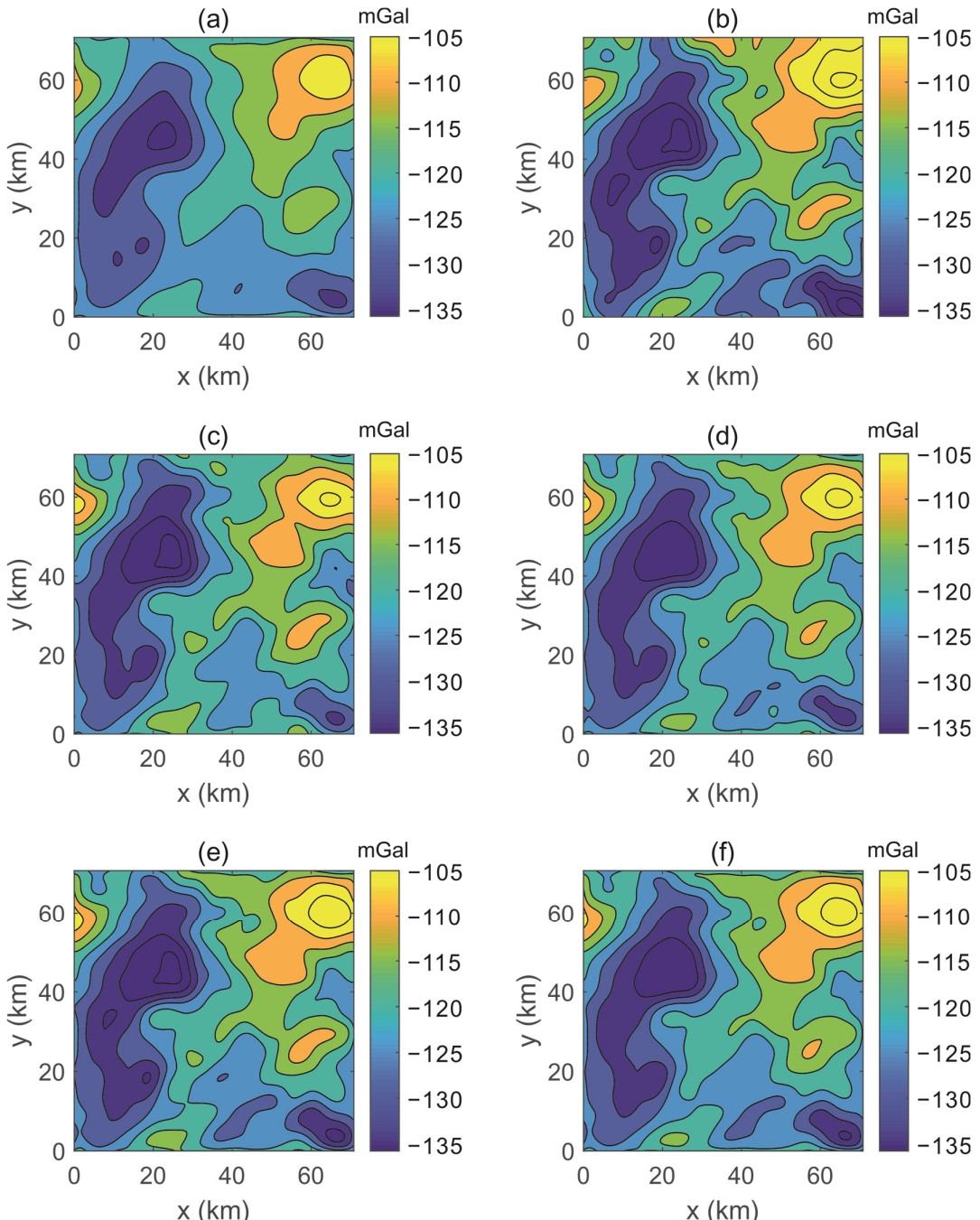

**Figure 8.** Both the observed airborne gravity anomaly and the observed vertical derivative are used to carry out downward continuation in the Nechako basin, with a downward continuation depth of 2000 m. (**a**) The gravity anomaly to be downward continued, which is obtained from the measured airborne gravity anomaly by upward continuation to 2200 m. (**b**) The observed gravity anomaly at the height of 200 m, which is taken as the reference gravity anomaly. (**c**) The downward continuation of (**a**) by the Milne method. (**d**) The downward continuation of (**a**) by the Milne–Simpson predictor-corrector method. (**e**) The downward continuation of (**a**) by the Adams–Bashforth method. (**f**) The downward continuation of (**a**) by the Adams–Bashforth–Moulton predictor-corrector method.

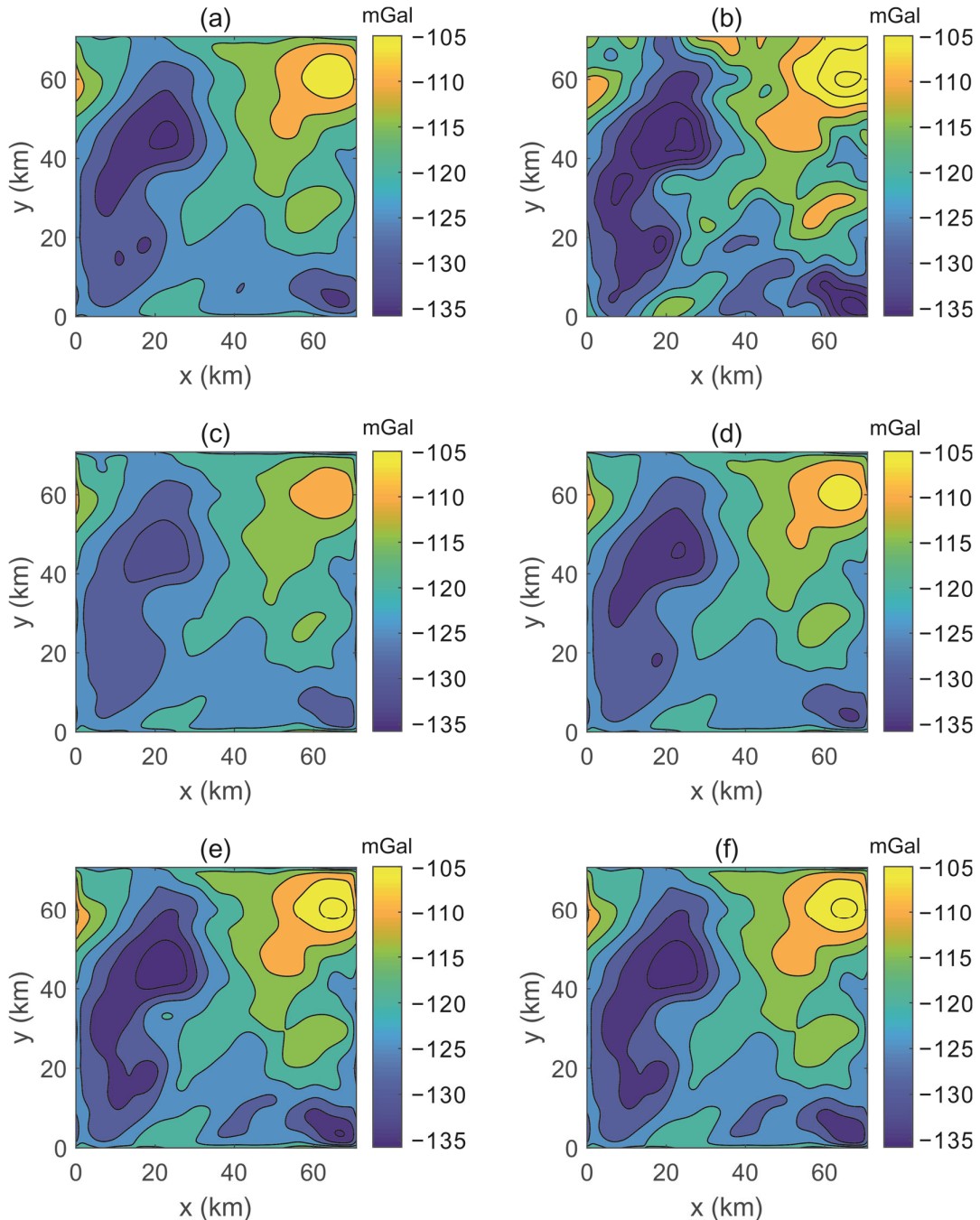

**Figure 9.** Only the observed airborne gravity anomaly is used to carry out downward continuation in the Nechako basin, with a downward continuation depth of 2000 m. (**a**) The gravity anomaly to be downward continued, which is obtained from the measured airborne gravity anomaly by upward continuation to 2200 m. (**b**) The observed gravity anomaly at the height of 200 m, which is taken as the reference gravity anomaly. (**c**) The downward continuation of (**a**) by the Milne method. (**d**) The downward continuation of (**a**) by the Milne–Simpson predictor-corrector method. (**e**) The downward continuation of (**a**) by the Adams–Bashforth method. (**f**) The downward continuation of (**a**) by the Adams–Bashforth–Moulton predictor-corrector method.

To better understand these four downward continuations based on numerical solutions of the mean-value theorem, we present RMS errors (Figure 10) between the observed airborne Bouguer gravity anomaly (Figure 8b or Figure 9b), which is taken as the reference anomaly, and the downward continuations (Figures 8c–f and 9c–f) with the downward continuation

depth from 400 m to 2000 m. Compared with the reference gravity anomaly (Figure 8b or Figure 9b), the downward continuation of the Adams–Bashforth method is the best (Figure 8e) with the input of the real vertical derivative, and that of the Adams–Bashforth–Moulton predictor-corrector method is the best without the real vertical derivative.

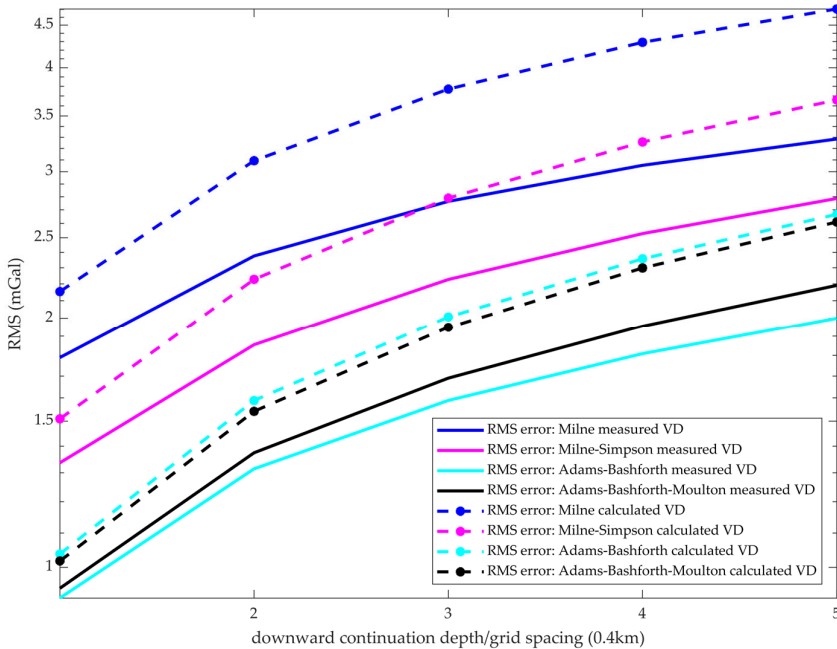

**Figure 10.** RMS errors between the observed airborne gravity anomaly, which is taken as the reference anomaly, and the downward continuations from the upward continuation gravity anomaly at the height of 200 m. Solid lines represent the real gravity anomaly and its vertical derivative (vertical derivative abbreviated as VD), which are obtained by upward continuation at the height of 2200 m as input. Dash lines represent only the real gravity anomaly obtained by upward continuation at the height of 2200 m as input. The blue lines are by the Milne method. The magenta lines are the Milne–Simpson predictor-corrector method. The cyan lines are the Adams–Bashforth method. The black lines are the Adams–Bashforth–Moulton predictor-corrector method.

## 4. Discussion

Behaviours of four downward continuation methods based on numerical solutions of the mean-value theorem coincide with truncated errors of corresponding mathematical formulae only under certain conditions, such as using theoretical gravity anomalies and their vertical derivatives at different heights from forward calculations as inputs; meanwhile, the downward continuation depth is no smaller than 4 m, as shown in the synthetic models of Figures 2b–e and 7a,b. Therefore, it is important to carry out careful tests, analyses, and comparisons to mathematical formulae in geophysics applications, even when using highly accurate mathematical methods. One cannot always assume that a mathematical formula will hold true under a variety of influencing factors when it is applied to geophysical calculations.

From Section 3.1.1, we can see that the RMS error of the Milne–Simpson predictor-corrector method is the smallest, and the RMS errors increase in the sequence of the Adams–Bashforth–Moulton predictor-corrector method, the Milne method, the Adams–Bashforth method, the integral iteration method, and the classic FFT method. RMS errors' sorting is in accordance with their truncation errors' first terms' behavior, which are $\frac{8h^5}{720}$ of the Milne–Simpson predictor-corrector method's truncation error $\frac{8h^5}{720}g^{(5)}(x, y, \zeta_{s,4})$, $\frac{19h^5}{720}$ of the Adams–Bashforth–Moulton predictor-corrector method's $\frac{19h^5}{720}g^{(5)}(x, y, \zeta_{am,4})$, $\frac{224h^5}{720}$ of the Milne method's $\frac{224h^5}{720}g^{(5)}(x, y, \zeta_{m,4})$, and $\frac{251h^5}{720}$ of the Adams–Bashforth method's $g^{(5)}(x, y, \zeta_{ab,4})$. So we may conclude here that though the last terms $g^{(5)}(x, y, \zeta_{s,4})$, $g^{(5)}(x, y, \zeta_{am,4})$, $g^{(5)}(x, y, \zeta_{m,4})$,

and $g^{(5)}(x, y, \zeta_{ab,4})$ in these truncation errors are different, for downward continuation with theoretical gravity anomalies and their vertical derivatives at different heights from forward calculations, the first terms of $\frac{8h^5}{720}$, $\frac{19h^5}{720}$, $\frac{224h^5}{720}$, and $\frac{251h^5}{720}$ are dominant for choosing a best method with the smallest RMS error.

As the difference between Sections 3.1.1 and 3.1.2 is the calculation factor of upward continuation, we can infer that the upward continuation applied to the inputs of downward continuations based on numerical solutions of the mean-value theorem will affect these methods. The RMS errors of the Adams–Bashforth–Moulton predictor-corrector method and the Adams–Bashforth method in Section 3.1.2 are almost the same as those in Section 3.1.1, but the RMS errors of the Milne–Simpson predictor-corrector method and the Milne method between these two parts change. This means that the upward continuation will affect little of the Adams–Bashforth–Moulton predictor-corrector method and the Adams–Bashforth method. Therefore, the Adams–Bashforth–Moulton predictor-corrector method may be the best choice for downward continuation if on the ground observation surface there is a gravity anomaly and its vertical derivative from forward calculations or observations.

RMS errors of the Adams–Bashforth–Moulton predictor-corrector method and the Adams–Bashforth method in Section 3.1.3 are smaller than the other methods; they increase compared with those of the Adams–Bashforth–Moulton predictor-corrector method and the Adams–Bashforth method in Sections 3.1.1 and 3.1.2, respectively. However, RMS errors of the Milne–Simpson predictor-corrector method and the Milne method in Section 3.1.3 decrease compared with corresponding ones in Section 3.1.2. Therefore, we may conclude here that the calculation of vertical derivatives will affect the downward continuation results. This indicates accurate measurements of vertical derivatives at the measurement height are important in data processing and anomaly interpretation in real exploration.

RMS errors of the Adams–Bashforth method and the Milne method are almost the same between Sections 3.1.3 and 3.1.4 but those of the Adams–Bashforth–Moulton predictor-corrector method and the Milne–Simpson predictor-corrector method change. This means that explicit methods such as the Adams–Bashforth method and the Milne method are less sensitive to noise than the presented two methods.

The Milne method and the Milne–Simpson predictor-corrector method are sensitive to upward continuation, while the Adams–Bashforth method and the Adams–Bashforth–Moulton predictor-corrector method are not as shown in Figures 3b–e and 7c,d. The vertical derivative at the measurement height calculated by the ISVD method affects the Adams–Bashforth method and the Adams–Bashforth–Moulton predictor-corrector method a lot, but the Milne method and the Milne–Simpson predictor-corrector method are affected little; however, the Adams–Bashforth–Moulton predictor-corrector method has the smallest RMS error shown in Figures 4b–e, 7e,f and 9. Hence, different factors have varying weights that affect the four methods of numerical solutions. Different methods should be considered concerning the downward continuation of different data.

RMS errors of these downward continuations based on numerical solutions vary smoothly and monotonously, and the result of the eighth-order Adams–Bashforth method is more accurate than that of the fourth-order Adams–Bashforth method used here [13]. Therefore, we can infer that higher than fourth-order methods based on numerical solutions would help to improve the accuracy of downward continuation, but further careful tests should be carried out in the future.

## 5. Conclusions

In this study, we presented two new methods for gravity anomaly downward continuation based on implicit expressions and their predictor-correctors, namely the Milne–Simpson predictor-corrector method and the Adams–Bashforth–Moulton predictor-corrector method. We evaluated the presented methods on both synthetic and real cases. The results demonstrated that new methods are valid for stable and accurate downward continuation

even at a downward continuation depth of up to eight depth intervals, and new methods produced better results than the integral iteration method.

Moreover, we found that the accuracy of four methods based on numerical solutions of the mean-value theorem, including two new methods presented in this study, can be affected by various calculation factors such as upward continuation, vertical derivative calculations, and noise during downward continuation and the affect weights of different factors are different for these four methods. This reminds us to be careful when applying methods in geophysical applications.

For most conditions, the new method of the Adams–Bashforth–Moulton predictor-corrector provides the best downward continuation; however, for the condition of theoretical (real) gravity anomaly and its vertical derivative known at the measurement height of 0 m from the forward calculation (observation), the Adams–Bashforth method is mainly better.

In the future, higher than fourth-order methods based on numerical solutions should be carried out to improve the accuracy of downward continuation and corresponding techniques of downward continuation on the undulate terrain or observation surface should be realized.

**Author Contributions:** Conceptualization, C.Z. and Q.L.; methodology, C.Z.; software, C.Z., P.Q., W.Z. and J.Y.; validation, P.Q. and J.Y.; formal analysis, C.Z., P.Q. and W.Z.; investigation, P.Q. and W.Z.; resources, Q.L.; data curation, C.Z.; writing—original draft preparation, C.Z.; writing—review and editing, C.Z., P.Q. and Q.L.; visualization, C.Z. and P.Q.; supervision, Q.L.; project administration, Q.L. and J.Y.; funding acquisition, C.Z., P.Q., Q.L. and J.Y. All authors have read and agreed to the published version of the manuscript.

**Funding:** This research was funded by the National Natural Science Foundation of China (Grant Nos. 41904122, 42004068, 42104136 and 92062108), the National Key R&D Program of China (Grant No. 2016YFC0600200), the China Geological Survey Project (Grant Nos. DD20221643, DD20190012, DD20230405 and DD20230299), the Science and Technology Projects in Guangzhou (Grant No. 202201011216), the "Macau Young Scholars Program" (Grant No. AM2020001), and the China Scholarship Council (Grant No. 202004180027).

**Data Availability Statement:** Not applicable.

**Acknowledgments:** The authors would like to thank David Gubbins, Keke Zhang, Longwei Chen, and Leyuan Wu for their support, discussions, and help in improving this paper and thank Tianxiang Ren for sharing the permission for MATLAB. The authors would also like to thank three anonymous reviewers for their constructive comments and suggestions.

**Conflicts of Interest:** The authors declare no conflict of interest.

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
