# Peer review of "Two New Methods Based on Implicit Expressions and Corresponding Predictor-Correctors for Gravity Anomaly Downward Continuation and Their Comparison"

_remotesensing, doi:10.3390/rs15102698_

Round 1
Reviewer 1 Report
Please see the report and annotated pdf file.

Author Response
Reviewer #1
General comments and suggestions:
In this manuscript the authors developed two methods based on implicit expressions to downward continuation for gravity data.
The manuscript needs some English work in terms of grammar, layout and flow. Please make sure that all figures and references are cited correctly in the text. The figures are of poor quality, and must be redrawn to meet academic standard. Some references of the text are not listed in the List of references. Please check.
Please find below are my specific comments. More comments are placed on the attached annotated pdf file - please see therein.
Reply:
Dear reviewer, thank you so much for your valuable comments. We have improved the language throughout the manuscript both by a native English scholar and an editing service. We redrew all the figures hopefully to meet the standard, and we also check all the references in the text and the corresponding reference list to make sure all the cited references relevant and no missing.
Specific comments
- The Examples and Comparison section needs rewrite and better organization.
Reply:
Thanks for your helpful suggestions. We have rewritten and reorganized the Examples and Comparison parts. As they are the whole part, we do not show any corrections here but details can be found in the ‘Track Changes’ version.
- Please provide the computation time for your calculations.
Reply:
Thank you for your suggestions. We have provided the computation time in Part 3.1.3:
‘We also estimate consuming times for six methods on a PC with i7 CPU at 2.9 GHz and 32.00 GB RAM, 0.0091 s, 0.0131 s, 0.0045 s, 1.2442 s, 0.0041 s, 1.2532 s for the classical FFT method, the integral iteration method, the Milne method, the Milne-Simpson predictor-corrector method, the Adams-Bashforth method, the Adams-Bashforth-Moulton predictor-corrector method.’
- Most of the figure captions are inappropriate. See for example, the caption of Figure 8. Please fix the figure captions of the entire paper.
Reply:
Thanks for your comments. We have fixed all captions of figures in our manuscript according to your suggestion.
- Please report the limitations of the proposed methods, and suggest avenues for future research.
Reply:
Dear reviewer, the limitations of our methods are reported in Part 4 and Part 5. Taking Part 5 as an example:
For most conditions, the new method of the Adams-Bashforth-Moulton predictor-corrector gives the best downward continuation, except for the condition of theoretical (real) gravity anomaly and its vertical derivative at the measurement height of 0 m from forward calculation (observation) and other input of downward continuation from upward continuation and vertical derivative calculations in synthetic model (real cases) with a downward continuation depth smaller than 4 m (5 times depth intervals), the Adams-Bashforth method are better.
For future research, we add them to Part 5:
In the future, higher than fourth-order methods based on numerical solutions should be carried out to improve the accuracy of downward continuation and correspongding techniques on deural measurement surface should be realized.
- Please check all figures and their citation. Please see above and the annotated text.
Reply:
Thank you so much for your comments. We have checked all the figures and their citations, and have replaced by new ones and corrected the wrong ones.
- Some references of the text are not listed in the List of references. Please check.
Reply:
Thanks again. We have checked all the references in the text and the corresponding reference list to make sure all the cited references relevant and no missing.
Reviewer 2 Report
In this research two new methods based on implicit expressions and corresponding predictor-correctors for gravity anomaly downward continuation and their comparison investigated. This is interesting study but need some revise.My comments are as follows:
- Summarize mathematical Formulas in section “Two Implicit Expressions for Gravity Anomaly”
- The quality of the figures is very low and unclear.
- Add a flowchart to implement the entire research.
- Possible future studies to be added.
-To add a discussion about the advantages and disadvantages of the proposed methods.
Reviewer 3 Report
The paper describes several algorithms of extrapolating gravity (or gravity anomaly) in the vertical direction, that is, up and/or down from where it was measured. The authors elaborate mathematical algorithms of the extrapolation, and then demonstrate it with a synthetic example and a real-world example. However, the presentation needs to be revisited for clarity and consistency.
1. The paper mentions measuring gravity and gravity anomaly interchangeably (lines 39-55: ‘A variety of gravity field data can be obtained from different altitudes’… ‘ground-based measurements of the gravity field’ … ‘Aerial gravity field measurements’ … ‘downward continuation of a gravity anomaly’ … ‘Downward continuation of gravity field data’)
So, what do you measure: gravity (the free fall acceleration?) or gravity anomaly (that is, the difference between the actual and expected values of gravity)? If the latter, then how this ‘expected’ value is determined?
2. All formulas in theoretical section 2.1 imply that the measurements are taken at four vertical levels with spacing h and then extrapolated through a distance h up or down. The operation can be repeated recursively, which would result in up/downward continuation through several intervals h.
However, in the example in 3.1, the authors “measure on the ground”. Please explain how you apply your formulas starting with measurements at a single elevation, on a single horizontal plane. Also please explain why the vertical reach of your ‘continuation’ depends on the spacing of the measurements in the horizontal plane (line 366: ‘downward continuation result is divergent when the depth is more than 2 times the point interval (2m) ), even though according to the theory, ‘continuation’ is done in the vertical direction only, with no horizontal dependancies.
3 Likewise, please list all heights at which the measurements were taken, in section 3.2 ‘Real data’.
Fig. 8 and 9 appear identical. The figures’ captions differ in their first sentences, but then explanations for panels a-f are identical again. So it’s not really clear what was done and what was obtained.
4. The Conclusions are confusing in the same way as the rest of the text. The authors state: “For both theoretical gravity anomaly and theoretical vertical derivative only at the height of the ground available, the Adams-Bashforth-Moulton predictor-corrector method yields the most accurate results when the downward continuation depth is bigger than four intervals” (lines 694-697). Which intervals? It would make sense if the vertical spacing h between the measurements were the scale of the ‘continuation’, but with the measurements ‘only at the height of the ground’, how do you determine those ‘four intervals’?
I can’t evaluate the soundness of the method until I know answers to the above questions.
Other comments:
5. ‘the gravity field data decays exponentially with the increase in height’ (line 48)
Please explain why it ‘decays exponentially’, contradictory to the common belief that the gravity field varies with the inverse square of the distance from the Earth center.
6. De-abbreviate ISVD
7. Lines 322-334 are duplicated in lines 337-349
8. Line 357: “depth of -8 m beneath the ground” - say either ‘elevation -8 m’ or ‘8 m below the ground’ but don’t use double negation (‘-‘ combined with ‘beneath’), here and elsewhere.
9. Figs. 1-4, 6 and 7 have unacceptably low resolution. What are units corresponding to the blurred numbers on the color-bars?
10. Line 423: “theoretically measured” ??? - a value is either theoretical, or measured
Round 2
Reviewer 1 Report
The authors made good effort and attempted to address my comments and concerns that I raised in the pdf file "report.pdf". However, it appears that the authors forgot to address my comments and concerns that I reported in the annotated pdf file of the original manuscript.
Please read closely and address the comments and concerns I made on each section of the original manuscript.
Please find attached is my earlier annotated file for your consideration.

Author Response
Reviewer #1
General comments and suggestions:
The authors made good effort and attempted to address my comments and concerns that I raised in the pdf file "report.pdf". However, it appears that the authors forgot to address my comments and concerns that I reported in the annotated pdf file of the original manuscript.
Please read closely and address the comments and concerns I made on each section of the original manuscript.
Please find attached is my earlier annotated file for your consideration.
Reply:
Dear reviewer, thank you so much for your comments. We are sorry for missing your comments in ‘the annotated pdf file of the original manuscript’. We will try to reply to and refine our manuscript to your comments and suggestions with a total number of 37 in our original manuscript carefully on this round of revision.
Additionally, we would like to ask for your forgiveness to explain that in Review Report-Round1, there is no ‘the annotated pdf file of the original manuscript’ on our side of the review report system.

In Review Report-Round1, there is only one ‘peer-review-28162331.v1.pdf’ file with only one page:

In Review Report-Round2, there is another one ‘peer-review-29234558.v1’ file of the annotated pdf file of the original manuscript:

- What does ISVD stand for? Please indicate.
Reply:
Thanks for your comment. ISVD stand for ‘integrated second vertical derivatives’ and we have de-abbreviated it. The details about the ISVD method can refer to
- Fedi, M.; Florio, G. A Stable Downward Continuation by Using the ISVD Method. Geophysical Journal International 2002, 151 (1), 146–156.
- Suggesting the following two relevantreferences:
Mehanee, S., 2022, A new scheme for gravity data interpretation by a faulted 2-D horizontal thin block: Theory, numerical examples and real data investigation, IEEE Transactions on Geoscience and Remote Sensing, Volume 60, page 1-14., doi: 10.1109/TGRS.2022.3142628.
Mehanee, S., 2022, Simultaneous joint inversion of residual gravity and self potential data measured along profile: Theory, numerical examples and a case study from mineral exploration with cross validation from electromagnetic data. IEEE Transactions on Geoscience and Remote Sensing, Volume 60, page 1-20. 10.1109/TGRS.2021.3071973.
Reply:
Thank you for your suggestions. We have added them to the reference and rewritten the order of the following references in the text.
- Please indicate page number and chapter number of this book. --original [6], --new [10]
Reply:
Thanks for your comment. We have added the corresponding chapter and pages to the reference, and the chapter is 12 from pages 319-320.
- The Fast Fourier Transform (FFT) .....
Reply:
Dear reviewer, thank you so much for this suggestion. We have changed it as per your suggestion and also change the ‘ISVD (integrated second vertical derivatives)’ into ‘integrated second vertical derivatives (ISVD)’ at their first appearing time.
- The rationale needs improvement.‘The Milne method and Adams-Bashforth method based on numerical solutions of the mean-value theorem using measured vertical derivative are stable, accurate and with deep continuation depth [10-12]. But these two methods’ truncation errors are not small enough compared to other methods based on numerical solutions of the mean-value theorem’.
Reply:
Thanks for your comment. We have improved this as the following:
The Milne and Adams-Bashforth methods based on numerical solutions of the mean-value theorem using the observed vertical derivative of the first order, simply called vertical derivative herein, at the measurement height, are reported to be stable, accurate and with deep continuation depth [14-16]. But these two methods’ truncation errors are not small enough mathematically compared to other methods based on numerical solutions of the mean-value theorem [17].
- The authors should highlight the benefits of the developed methods.‘for gravity downward continuation. As the advantages of downward continuations based on numerical solution methods are limited by the measured vertical derivative, we consider more widely application scenarios such as those without measured vertical derivative, while still maintaining stability, accuracy, and deep continuation depth. We introduce’.
Reply:
Thanks for your comment. We have improved this as the following:
for gravity anomaly downward continuation. As the advantages of previous downward continuations based on numerical solution methods are limited by the observed vertical derivatives at the measurement height, we consider wider application scenarios using the calculated vertical derivatives, such as those without observed vertical derivatives, while maintaining stability, accuracy, and deep continuation depth. We introduce
- 7. This shouldbe placed in the Conclusion section.This is a finding can not be here. ‘The results show that new methods can provide stable, accurate and deep-depth downward continuations. Factors such as upward continuation, vertical derivative calculation and noises affect the four methods of downward continuations of numerical solution methods, but to different extents. Overall, the downward continuation methods base on numerical solutions of the mean-value theorem perform better than the integral iteration method.’.
Reply:
Thanks for your comment. We have changed this part as the following:
Factors, such as calculated gravity anomalies and their vertical derivatives at non-measurement heights above the observation by upward continuation, calculated vertical derivatives of gravity anomalies at the measurement height by the ISVD method, and noises, do affect the four numerical solution-based downward continuation methods, but to different extents. Overall, the downward continuation methods base on numerical solutions of the mean-value theorem perform better than the integral iteration method and the newly presented Adams-Bashforth-Moulton method predictor-corrector method is a better choice than the other three numerical solution-based ones in general.
- Please add a few statementsunder the methods to explain the coming information. ‘2. Methods’.
Reply:
Thanks for your suggestions. We have added some to explain the coming information:
For understanding methods, we first present general expressions of numerical solutions of the mean-value theorem for gravity anomalies. Second, we recall the explicit Adams-Bashforth and explicit Milne Expressions for gravity anomaly downward continuation. Finally, we derive and present two implicit expressions of Adams-Moulton and Simpson, and their predictor-corrector methods of Adams-Bashforth-Moulton and Milne-Simpson for downward continuation.
- 9. Please add a few statementsunder this sub-sectionto explain the coming information.
Reply:
Thanks for your comment. We have added some under the part of ‘2. Methods’ the reply 8.
- 10. where the coefficients ......
Reply:
Dear reviewer, thank you so much for this suggestion. We have changed it as per your suggestion.
- 11. please delete.
Reply:
The same as the above reply.
- 12. slightly. Howit is slightly different.Please clarify.
Reply:
Thanks for your comment. We have changed this sentence as:
The expression (2) is slightly different in symbols and their meanings from the recurrence relation of multistep methods of equation 3.41 in [17]
- 13. where .
Reply:
Thanks for your comment but we choose not to change it. As the end of the formula (5) is a ‘.’, this should be a capital ‘Where’ at the beginning of a sentence, not ‘where’.
- Please add a few statementsunder this subsection to explain the coming information.
Reply:
Thanks for your comment. We have added some under the part of ‘2. Methods’ the reply 8.
- This density contrast is very small. Do you mean gm/cm^3? Please check.
Reply:
Dear reviewer, thank you so much for this suggestion. Yes, it should be g/cm^3. We have changed all corresponding as per your suggestion.
- This density contrast is very small. Do you mean gm/cm^3? Please check.
Reply:
The same as reply 15.
- This density contrast is very small. Do you mean gm/cm^3? Please check.
Reply:
The same as reply 15.
- The figure is hazy. Please fix (re-plot).
Reply:
Dear reviewer, thank you for your comment.
We also notice the problem of the low resolution of figures in our pdf file. As larger pdf with high-resolution figures can not be uploaded in ‘PDF File’, we upload an another .zip file inside individual figures, a docx file with high-resolution figures. It is our pleasure to have your kind help to download the .zip file from ‘Manuscript File’ to check the figure quality. We hope this can meet the requirement of the resolution of figures for publication and also the system of remote sensing submission.
- This is a repeat. This was said above. Please delete.
Reply:
Thank you for your suggestions. We have deleted it in the previous revision.
- 20. The photos of Figure 2 are hazy and quite poor. Please fix all of them. The font of each photo is tiny. Please use larger font. What arethe differences between Figures 2, 3, 4, 5 and 6? This should be made clear before discussing Table 1. The x and y-labels are too tiny. Please make things larger.
Reply:
Thanks for this comment. The same as reply 15 about the figures.
The differences between Figures 2, 3, 4, 5 and 6, We have clarified them in the previous revision. As:
For Figure 2:
3.1.1. Downward continuation with theoretical gravity anomalies and their vertical derivatives at different heights from forward calculations.
For Figure 3:
3.1.2. Downward continuation with the theoretical gravity anomaly and its vertical derivative at the measurement height of 0 m from forward calculations.
For Figure 4 and Figure 5:
3.1.3. Downward continuation with the theoretical gravity anomaly at the measurement height of 0 m from the forward calculation.
For Figure 6:
3.1.4. Downward continuation with the theoretical gravity anomaly at the measurement height of 0 m from the forward calculation with Gaussian white noises.
- 21. What arethe differences between Figures 2, 3, 4, 5 and 6?
Reply:
The same as reply 20.
For the differences between Figures 2, 3, 4, 5 and 6, we have clarified them in the previous revision.
- 22. What arethe differences between Figures 2, 3, 4, 5 and 6?
Reply:
The same as reply 20.
For the differences between Figures 2, 3, 4, 5 and 6, we have clarified them in the previous revision.
- 23. What arethe differences between Figures 2, 3, 4, 5 and 6?
Reply:
The same as reply 20.
For the differences between Figures 2, 3, 4, 5 and 6, we have clarified them in the previous revision.
- 24. What arethe differences between Figures 2, 3, 4, 5 and 6?What depth is these results for?
Reply:
The same as reply 20.
The differences between Figures 2, 3, 4, 5 and 6, we have clarified them at previous revision.
About the depth, we have clarified them in the previous revision. Like ‘All downward continuation depths are 8 m (8 times depth interval) which is at the measurement height of -8 m.’
- What depth is these results for?
Reply:
About the depth, we have clarified them in the previous revision. Like ‘All downward continuation depths are 8 m (8 times depth interval) which is at the measurement height of -8 m.’
- What arethe differences between Figures 2, 3, 4, 5 and 6?
Reply:
The same as reply 18.
- 27. Figures are hazy and poor quality? Please fix all of them?
Reply:
The same as reply 20.
The differences between Figures 2, 3, 4, 5 and 6, we have clarified them at previous revision.
- 28. This section needs to be organized better and re-worded.
Reply:
Thanks for your comment. We have improved this as the following:
‘To verify the actual downward continuation by the Milne-Simpson predictor-corrector method and the Adams-Bashforth-Moulton predictor-corrector method proposed in this study, we use the gravity anomaly from the airborne measurement at the height of 200 m over the Nechako Basin area of Canada. Both the airborne Bouguer gravity anomaly (Figure 8b) and its vertical derivative are available in this area. The grid spacings of the gravity anomaly and its vertical derivative are 400 m. For testing, we upward continue the gravity anomaly (Figure 8b) and its vertical derivative from their measurement height of 200 m to a height of 2200 m (equivalent to 5 spacing intervals) and consider this upward continuation gravity anomaly (Figure 8a) at the height of 2200 m as the one to be downward continued.
First, we use both the real gravity anomaly (Figure 8a) and the real vertical derivative which are obtained by upward continuation at the height of 2200 m as input to downward continue by the Milne method, the Milne-Simpson predictor-corrector method, the Adams-Bashforth method, and the Adams-Bashforth-Moulton predictor-corrector method, respectively. Results are shown in Figures 8c to 8f. The two presented methods can give stable and accurate downward continuations.’
- 29. what was the measuring height?
Reply:
Thanks for your comment, we have clarified this in the previous revision.
‘we use the gravity anomaly from the airborne measurement at the height of 200 m over the Nechako Basin area of Canada.’
- Figure 8a should be citedbefore Figure 8b.
Reply:
Thanks for your comment. As the gravity anomaly in Figure 8b is real measured and the value in Figure 8a is the upward continuation result from the real measured gravity anomaly in Figure 8b. Figure 8b is cited before Figure 8a.
- Figure 8a should be citedbefore Figure 8b.
Reply:
Thanks for your comment. As the gravity anomaly in Figure 8b is real measured and the value in Figure 8a is the upward continuation result from the real measured gravity anomaly in Figure 8b. Figure 8b is cited before Figure 8a.
- 32. This is confusing because you said you will use the results of Figure 8a for downward continuation.
Reply:
Thanks for your comment. We use the upward continuation result in Figure 8a to downward continue. The upward continuation result in Figure 8a is from the real measured gravity anomaly in Figure 8b. The real measured gravity anomaly in Figure 8b is like ‘reference true values’. There are no true values in real cases, so we take the measured gravity anomaly as them.
- 33. Photos are poor quality.x(km) should be x (km). Please check the entire figures.
Reply:
Thank you for this comment. The same as reply 20. And we also changed the ‘x (km)’.
- 34. What is the point of doing these calculations? How is this different from the calculations of Figure 8? Please clarify and explain for the reader of your paper.
Reply:
Thanks for your comment. The input of results in Figure 8 includes ‘gravity anomalies’ and ‘measured vertical derivatives’; while the input of results in Figure 9 includes ‘gravity anomalies’ and ‘calculated vertical derivatives’ which are calculated from ‘gravity anomalies’ by the ISVD method.
We have improved this as the following:
‘To widely illustrate applications of the proposed methods, we carry out the downward continuations only using the observed gravity anomaly at the height of 2200 m. The corresponding results of the Milne method, the Milne-Simpson predictor-corrector method, the Adams-Bashforth method, and the Adams-Bashforth-Moulton predictor-corrector method are shown in Figures 9c to 9f. Also, reasonable results are obtained by the presented methods.
To better understand these four downward continuations by numerical solutions, we present their RMS errors in Figure 10. Compared with the reference gravity anomaly in Figures 8b or 9b which are the observed airborne Bouguer gravity anomaly, the downward continuation of the Adams-Bashforth method with real vertical derivative is the best and that of the Adams-Bashforth-Moulton predictor-corrector method without real vertical derivative are the best.’
- 35. How is this different from Figure 8? Photos are poor quality. x(km) should be x (km). Please check the entire figures.
Reply:
Thanks for your comment. The same as reply 33 and reply 34.
- 36. Please remove the dots and re-write the discussion as one paragraph or as subsection if the discussion requires that.
Reply:
Thanks for your suggestion. We have improved this part.
- 37. Please discuss these factors, and elaborate on them.
Reply:
Thanks for your comment. We have improved the conclusion and elaborated on them both in the conclusion part and the discussion part and partly the model part.
We have uploaded a word for the reply to the review report. Please see the attachment.

Reviewer 3 Report
The authors have addressed my previous comments, that is, the presentation clarity has improved. One might argue that taking a gravity measurement at 200 m above the ground, re-calculating it to the height of 2200 m (‘upward continuation’), then playing it back to 200 m (‘downward continuation’) and recovering the starting value is not a convincing proof of physical soundness of either ‘continuation’: ‘’upward ’and ‘downward’ continuations have simply undone each other, regardless of their physical validity. But at least, the presentation is clear enough for a reader to make their judgement.
I still have a concern about figures quality: axes labels and texts inside the plots are unreadable in most figures - at least, in the pdf which I’m reviewing - the reason why I could not select ‘accept in present form’.
Author Response
Reviewer #3
General comments and suggestions:
The authors have addressed my previous comments, that is, the presentation clarity has improved. One might argue that taking a gravity measurement at 200 m above the ground, re-calculating it to the height of 2200 m (‘upward continuation’), then playing it back to 200 m (‘downward continuation’) and recovering the starting value is not a convincing proof of physical soundness of either ‘continuation’: ‘’upward ’and ‘downward’ continuations have simply undone each other, regardless of their physical validity. But at least, the presentation is clear enough for a reader to make their judgement.
I still have a concern about figures quality: axes labels and texts inside the plots are unreadable in most figures - at least, in the pdf which I’m reviewing - the reason why I could not select ‘accept in present form’.
Reply:
Dear reviewer, thank you so much for your understanding, recommendation and help.
We also notice the problem of the low resolution of figures in our pdf file. As larger pdf with high-resolution figures can not be uploaded in ‘PDF File’, we upload an another .zip file inside are individual figures, a docx file with high-resolution figures. It is our pleasure to have your kind help to download the .zip file from ‘Manuscript File’ to check the figure quality. We hope this can meet the requirement of the resolution of figures for publication and also the system of remote sensing submission.

We have uploaded a word. Please see the attachment.

Round 3
Reviewer 1 Report
The authors addressed my comments and concerns. I think the revised manuscript is ready for acceptance.